# Tectonic and climatic implications of the Aleutian Arc initiation ≥56 million years ago

K. Hoernle [1] ✉, B. Jicha [2], M. Portnyagin [1], S. Zahirovic [3], D. Müller [3], F. Hauff [1], C. Timm [1], T. W. Höfig[4], G. Yogodzinski [5], M. Guillong [6], C. Berndt [1], D. Savelyev [7], R. Bezard [8] & B. Baranov [9]

The timing and origin of Aleutian subduction initiation remain poorly constrained, yet they are central to understanding late Paleocene to early Eocene plate tectonic reorganization in the Pacific and its possible climatic consequences. Here, we use geochronologic and geochemical data obtained on western Aleutian arc samples from four basal submarine sequences, spanning ~700 km, to constrain Aleutian subduction initiation to ≥56 Ma. Early forearc lavas have similar compositions to forearc basalts erupted during the initial stages of Izu-Bonin-Mariana subduction in the western Pacific. Collision of the Olyutorsky Arc with Kamchatka-Koryak margin and subduction of the Izanagi-Pacific Ridge are likely to have triggered Aleutian subduction initiation and a change in absolute Pacific plate motion from WNW to N between 57 and 55 Ma, as shown with a GPlates model. Our study shows that Aleutian subduction initiation is a key event at the beginning of a major ~10 Myr plate reorganization in the circum-Pacific ending with Hawaii-Emperor-Bend formation. These tectono-magmatic events may have contributed to contemporaneous global climatic events in the late Paleocene and early Eocene.

Establishing the timing of Aleutian Arc initiation is critical for understanding the origin of Aleutian subduction and the broader early Cenozoic reorganization of plate tectonics in the Pacific Basin. This reorganization includes subduction initiation in the Izu–Bonin–Mariana and Tonga systems of the western Pacific and tectono-magmatic changes along the North American Cordilleran margin from Alaska through Canada to the northwestern United States. The Pacific Plate currently subducts beneath the nearly 4,000-km-long Aleutian–Alaska Arc, which extends from the Gulf of Alaska to the Kamchatka Trench (Fig. 1a), yet the timing and cause of Aleutian subduction zone initiation (SZI) remain controversial. Although the oldest published K/Ar ages from the Aleutian Arc extend back to 55 Ma, they are imprecise and provide only weak

temporal constraints because of their large uncertainties (e.g., 55.3 ± 6.7 Ma[1]; 45 ± 5 Ma[2]). Moreover, these ages were not reproduced by subsequent $^{40}Ar/^{39}Ar$ analyses of samples from the same regions[3]. The oldest precise $^{40}Ar/^{39}Ar$ ages, 46.31 ± 0.91 Ma[4] and 47.70 ± 0.11 Ma[5], come from two tholeiitic lavas recovered from near the base of Murray Canyon in the central Aleutians at water depths greater than 3,000 m. These data provide a minimum age of 48 Ma for Aleutian SZI and have therefore been used to suggest a possible link to formation of the Hawaiian–Emperor Bend at 50–47 Ma[5–8]. Published reconstructions, however, propose a wide range of initiation ages, from 55–46 Ma, despite the limited reliable radioisotopic, magnetic, and paleontological evidence supporting ages older than ~ 48 Ma.

[1]Research Division 4, GEOMAR Helmholtz Center for Ocean Research Kiel, Kiel, Germany. [2]Department of Geoscience, University of Wisconsin-Madison, Madison, Wisconsin, USA. [3]EarthByte Group, School of Geosciences, The University of Sydney, Sydney, Australia. [4]Projektträger Jülich, Forschungszentrum Jülich GmbH, Rostock, Germany. [5]School of Earth, Ocean, and Environment, University of South Carolina, Columbia, SC, USA. [6]Department of Earth and Planetary Sciences, ETH Zurich, Zurich, Switzerland. [7]Institute of Volcanology and Seismology FEB RAS, Petropavlovsk-Kamchatsky, Russia. [8]Laboratoire de Planétologie et Géosciences, Nantes Université, Nantes, France. [9]Shirshov Institute of Oceanology, Russian Academy of Sciences, Moscow, Russia. ✉e-mail: khoernle@geomar.de

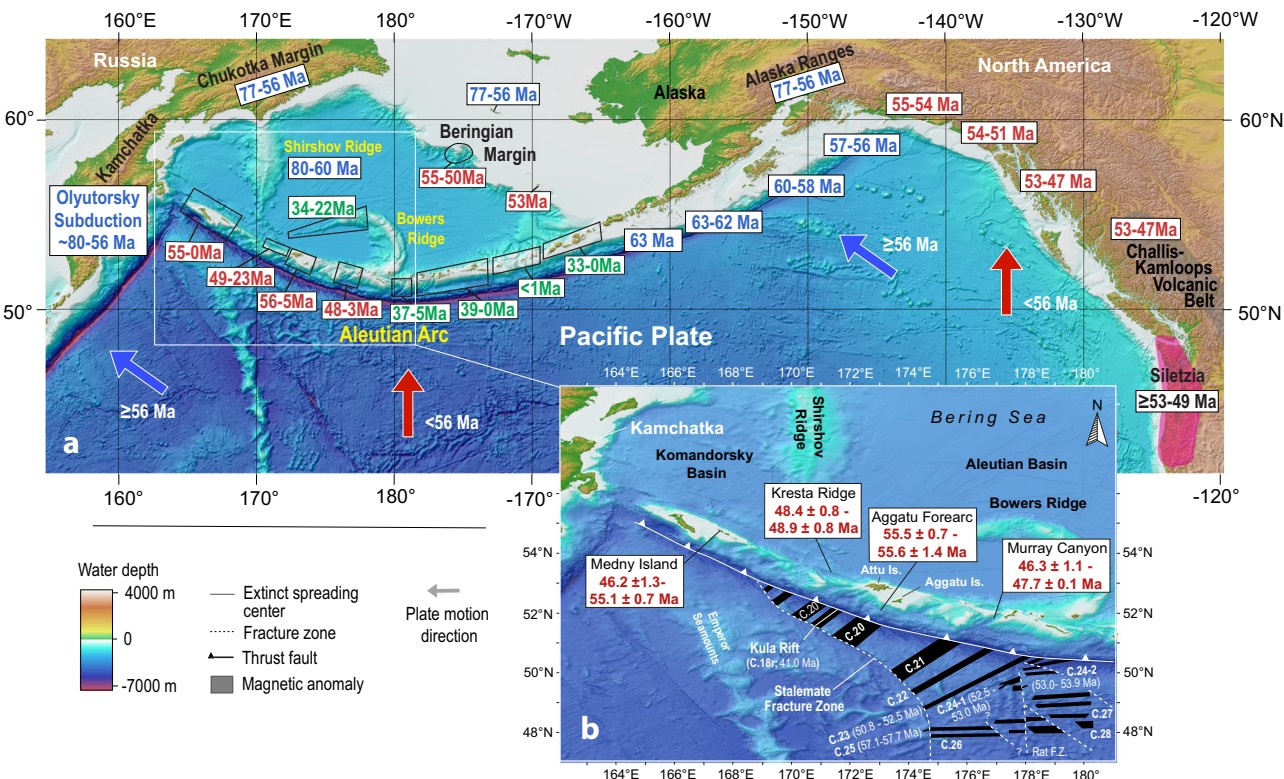

**Fig. 1 | Maps of North Pacific region showing ages of arc volcanism. a** Initiation of the western Aleutian arc at ≥ 56 Ma coincides with a change in Pacific Plate motion from WNW (blue arrows corresponding to arc ages in blue) to N (red arrows and red ages). K/Ar, Ar/Ar and U/Pb zircon ages of the oldest credibly dated Paleocene-Eocene magmatism are given in million years (Ma). **b** Inset: Range of the oldest ages from the western Aleutians. Magnetic anomalies on the seafloor south of the western Aleutian Arc from ref. 21 are also shown. Bathymetric data are from the GEBCO 2023 Grid, available at DOI 10.5285/f98b053b-0cbc-6c23-e053-6c86abc0af7b. Additional sources of age data[2–5,17,28,31–33,36,37,46–48,50,89–91]. The oldest ages from regions that are < 40 Ma are in green.

A range of scenarios have been proposed for the timing and cause of Aleutian SZI, as well as the formation of the Shirshov and Bowers Ridges. Combining two previous general models, Scholl[9] proposed that the accretion of the Olyutorsky Arc to the Kamchatka-Koryak margin and the extrusion of western Alaska crust caused the Beringian margin to jump southward, forming an offshore continuation of the Alaska Peninsula subduction zone at ~ 50 Ma. In this model, SZI captured a piece of the Pacific/Kula Plate to form the Aleutian Basin, whereas westward extrusion of western Alaska crust caused buckling of the Aleutian Basin crust and offshore formation of the Shirshov and Bowers Ridge subduction zones at ~45 Ma. Using constraints from continental-margin geology, seismic tomography, and paleomagnetism, Domeier et al. ref. 10. argued that two intraoceanic subduction zones with opposite polarity, Olyutorsky and Kronotsky, were required to explain the kinematic history of the North Pacific Ocean from Late Cretaceous to the Paleocene. They proposed that early Eocene arc-continent collision terminated southeast-dipping subduction beneath the Olyutorsky Arc, which triggered Aleutian SZI at > 46 Ma. They found it puzzling, however, that north-dipping subduction under the Aleutian arc started at Hawaiian–Emperor Bend time, since Aleutian subduction should have produced a northward pull, not the observed shift from northward to northwest motion. Vaes et al.[11]. favored Aleutian SZI at 50 Ma and also linked it to the collision of the Olyutorsky Arc with the Kamchatka–Koryak margin. They suggested that subduction initiated along a transform fault or fracture zone, as also proposed by refs. 9,12, separating the northern and southern Olyutorsky Arc off Kamchatka. Similar to ref. 9, they show subduction initiating under the eastern Aleutian arc with a strike-slip fault connecting the western portion of the Aleutian arc to the Kamchatka-Koryak margin. To explain the roughly N-S magnetic anomalies on the Aleutian basin seafloor, they proposed that the captured crust forming the basin

seafloor originated through Olyutorsky backarc spreading from ~ 85 to 50 Ma. They further suggested that the Shirshov Ridge split off the Olyutorsky Arc between 60 and 50 Ma and that collision of the Kronotsky Arc with the Aleutian Arc knocked out a piece of the western Aleutian Arc to form the Bowers Ridge at ~ 35 Ma. Finally, they interpret the Komandorsky Block to be a piece of the Kronotsky Arc accreted to the westernmost Aleutian arc in the Miocene. Crameri et al. ref. 13. developed a comprehensive database of subduction zone initiation events and classified Aleutian SZI as an example of polarity reversal, in which subduction stepped offshore and reversed direction after occlusion of Olyutorsky subduction upon collision with the Kamchatka-Koryak margin at ~ 53 Ma. Most recently, Stern et al. ref. 14. proposed Aleutian SZI at 54–51 Ma and discussed it in the context of Aleutian marginal basin formation. They considered two end-member models for Aleutian Basin formation: backarc spreading and ocean-plate capture. In their backarc spreading model, the Shirshov and Bowers Ridges originally formed the Beringian margin forearc and were rifted away by N-S-oriented spreading centers, in order to explain the roughly N-S magnetic lineations in the Aleutian Basin. Accordingly, arc volcanism on the Shirshov and Bowers Ridges would have begun in the early Eocene, but the basement would have been older forearc material from Beringian margin subduction. In their plate capture scenario, the Bowers Ridge formed in the late Eocene to Oligocene and the Shirshov Ridge in the Miocene. In both models, the Aleutian Arc formed through westward propagation of Alaska Peninsula subduction, with the westernmost part of the arc being connected to Kamchatka subduction through a transform fault, similar to the models of[9,11,12]. Collectively, these models imply either that the westernmost Aleutian Arc should be younger than the eastern Aleutians or that the westernmost Aleutians represent exotic terranes from another arc. Overall, previously proposed Aleutian SZI ages range from 55 to 46 Ma

and models fall into two broad groups: 1) collision of the Olyutorsky arc with Kamchatka, leading to a seaward subduction jump and polarity reversal, or 2) Beringian arc splitting and backarc spreading.

The western Aleutians provide a rare opportunity to test these competing models because tectonic segmentation and faulting expose older basement sequences. Although subduction is presently nearly orthogonal beneath the eastern Aleutians, it becomes increasingly oblique westwards. This transition has fragmented the central and western Aleutians into clockwise-rotating blocks and opened large transverse tear canyons that cut the arc massif, such as Murray Canyon. In the far western Aleutians, these blocks give way to major right-lateral strike-slip faults, including the Bering-Kresta fault system[15]. The south-facing Kresta Ridge fault scarp, located in the backarc, exposes an ~ 2000 m vertical section of igneous basement. Key basement sections at Murray Canyon ( ~ 177 °E), the Agattu forearc ( ~ 173 °E) and Kresta Ridge ( ~ 171 °E) were sampled by dredging during the German RV *Sonne* SO249 cruise[16], as part of the German-Russian-American collaborative project BERING. Samples were collected from the Komandorsky Series, the oldest stratigraphic unit on Medny Island ( ~ 167 °E), reported to have Upper Paleocene foraminifera at the base of the sequence[2,17], within the German-Russian collaborative project KALMAR.

Here, we present U/Pb and Ar/Ar geochronology and/or whole-rock geochemical data for lavas and granodiorites from the four submarine basement sequences spanning the western Aleutian Arc. We show that Aleutian SZI occurred at ≥ 56 Ma, significantly earlier than most previous estimates. This timing places Aleutian SZI near the onset of a ~ 10 Myr circum-Pacific plate reorganization and suggests a possible link to contemporaneous global climatic events, including the Paleocene–Eocene Thermal Maximum and the Eocene Hothouse[18].

## Results
### Age and chemistry constrain Aleutian subduction initiation

Zircons from five granodiorite samples from the Kresta Ridge produced U-Pb ages ranging from 48.44 ± 0.83 Ma to 48.93 ± 0.82 Ma (sample DR40-2), giving a mean age of 48.66 ± 0.74 (Fig. 1b, Table 1, Supplementary Table 1 and Fig. 1). Granodiorite sample DR40-2 provides a minimum age for an enclosed basaltic lava xenolith (DR40-2xeno), which must be from the base of the Aleutian arc and thus is likely to be several million years older than the granodiorite. Three different tholeiitic basalts produced indistinguishable $^{40}Ar/^{39}Ar$ plateau ages of 55.59 ± 1.41 Ma and 55.49 ± 0.72 Ma (from the Agattu forearc) and 55.06 ± 0.19 Ma (from Medny Island, 450 km from the Agattu forearc) (Fig. 1b, Table 2, Supplementary Table 2 and Fig. 2). Five other Medny lavas demonstrate continuous activity from 52.20 ± 0.6 Ma to 46.2 ± 1.3 Ma. Therefore, our data show that three

locations in the Aleutians extend to much older ages ( ≥ 48.7–55.6 Ma) than published ages from Murray Canyon.

Samples from Agattu, Murray (including previously dated oldest samples) and Kresta localities primarily consist of tholeiitic basalt to basaltic andesite and granodiorite, whereas lavas from Medny consist of calc-alkaline basalts and rhyolite (Supplementary Table 3,4). Due primarily to their high $TiO_2$ (0.7-1.9 wt%), none of the samples have boninitic compositions (i.e., $SiO_2 > 52$ wt%, $MgO > 8$ wt% and $TiO_2 < 0.5$ wt%[19]). On a diagram of relatively fluid immobile incompatible elements normalized to average normal-mid-ocean-ridge basalt (N-MORB), the Agattu samples straddle the average N-MORB composition, but show distinct deviations characteristic of subduction zone signatures (Fig. 2a). Samples from all localities show Nb-Ta depletions relative to average N-MORB and most relative enrichment in Ba, Th, Pb and light rare earth elements (LREE) (Fig. 2b). Initial Pb isotope ratios (Supplementary Table 4) form good to excellent positive linear correlations with incompatible trace element ratios, e.g., $^{207}Pb/^{204}Pb(t)$ versus Th/La ($R^2 = 0.94$; Fig. 3a), Th/Nb (0.94), Th/Yb (0.94); La/Yb (0.90), Sr/Yb (0.82), Pb/Yb (0.80), Pb/Nd (0.72) and Ba/Yb (0.62). Initial Pb isotope ratios also show excellent correlations with each other (e.g., $^{207}Pb/^{204}Pb(t)$ vs $^{208}Pb/^{204}Pb(t)$, $R^2 = 0.98$, Fig. 3b) and with $^{143}Nd/^{144}Nd(t)$ ($R^2 = 0.95$; Fig. 3c), increasing in the sequence Agattu, Murray, Kresta, Medny. Similar arc-type geochemical compositions and ages from each of the four locations and the excellent correlations formed by the data confirm that the recovered samples represent local arc rock units and not glacial dropstones. The aforementioned correlations can be explained by two-component mixing of a depleted (upper mantle) with an enriched (North Pacific sediment) component.

The oldest lavas in the Aleutians (56–41 Ma) come from four localities spanning the western half of the arc. These localities not only have overlapping age ranges, but also form good to excellent correlations on trace element and isotope diagrams, supporting derivation from the same subduction system. Therefore, it is unlikely that the westernmost Aleutians (Komandorsky Block) represent an accreted part of the Kronotsky Arc (e.g.,[11,20]), an island arc that formed at mid-latitudes and migrated northwestwards until it collided with Kamchatka near the Aleutian junction in the late Miocene to Pliocene. Moreover, continuous volcanism from 55-8 Ma (younger ages from ref. 2) is not consistent with the Komandorsky Block representing a piece of the Kronotsky Arc, since the Kronotsky Arc presumably shut off at ≥ 42 Ma and was not accreted until mid-Miocene[11,20]. The close similarity in age and geochemistry of lavas from the four western Aleutian localities indicates derivation from the same arc, which is most likely to be the incipient Aleutian Arc, in particular samples from Kresta Ridge forming the northern scarp of the Bering-Kresta fault.

**Table 1 | Summary of in-situ LA-ICP-MS zircon U-Pb ages from Kresta Ridge granodiorites, western Aleutians**

| Sample # | Concordant data / all data" | Apparent age* Ma | Uncertainity** (Ma) ± 2σ | Uncertainity^ (Ma) ± 2σ | MSWD^^ |
|---|---|---|---|---|---|
| SO249 DR40-1 | 49/55 | 48.51 | 0.23 | 0.76 | 0.97 |
| SO249 DR40-2 | 50/50 | 48.93 | 0.37 | 0.82 | 2.7 |
| SO249 DR40-3 | 53/53 | 48.49 | 0.39 | 0.83 | 2.6 |
| SO249 DR40-6 | 50/52 | 48.44 | 0.41 | 0.83 | 2.7 |
| SO249 DR40-8 | 53/57 | 48.81 | 0.36 | 0.82 | 2.4 |
| Mean | 255/267 | 48.66 | 0.16 | 0.74 | 2.3 |

" Number of analyses.

\* Mean 206Pb/238U age not corrected for common Pb. Discordant ages were excluded.
See supplementary data tables and method description for details.

\*\*Weighted mean age uncertainties

^ Propagated uncertainty is calculated by quadratic adding of uncertainty of ages and
1.5% external error of LA-ICP-MS measurements.

^^MSWD=Mean Standard Weighted Deviation.

**Table 2 | Summary of $^{40}$Ar/$^{39}$Ar age data from Agattu Forearc and Medny Island basalts and a Medny rhyolite from the western Aleutians**

| Sample # | Location | Material* | Expts | K/Ca | $^{40}$Ar/$^{36}$Ar$_i$ | | Isochron Age | | N | $^{39}$Ar% | MSWD | Plateau Age | |
|---|---|---|---|---|---|---|---|---|---|---|---|---|---|
| | | | Nr # | total | ± 2σ | | (Ma) ± 2σ | | | | | (Ma) ± 2σ | |
| K8-8 | Medny | grd | 1 | 0.011 | 298.1 | ± 6.2 | 46.3 | ± 2.7 | 18/19 | 99.5 | 0.75 | 46.16 | ± 1.31 |
| K8-11 | Medny | plag | 2 | 0.009 | 293.2 | ± 9.4 | 50.0 | ± 3.8 | 27/47 | 74.3 | 0.37 | 47.90 | ± 1.20 |
| K8-10 | Medny | grd | 1 | 0.020 | 295.9 | ± 5.4 | 50.9 | ± 5.4 | 26/40 | 63.2 | 0.51 | 48.22 | ± 1.10 |
| K8-9 | Medny | grd | 1 | 0.012 | 300.3 | ± 3.5 | 51.1 | ± 1.6 | 22/22 | 100.0 | 1.01 | 51.83 | ± 0.72 |
| K8-18 | Medny | grd | 2 | 0.027 | 298.9 | ± 6.7 | 52.0 | ± 2.3 | 26/37 | 80.9 | 1.29 | 52.19 | ± 0.58 |
| K8-15 | Medny | grd | 2 | 0.016 | 299.5 | ± 0.8 | 54.5 | ± 0.4 | 37/50 | 84.6 | 0.04 | 55.06 | ± 0.19 |
| SO249 DR51-6 | Agattu | grd/plag | 3 | 0.005 | 299.2 | ± 0.9 | 54.7 | ± 1.6 | 42/57 | 87.7 | 0.65 | 55.49 | ± 0.72 |
| SO249 DR51-9 | Agattu | grd | 3 | 0.003 | 299.1 | ± 6.5 | 55.6 | ± 6.6 | 52/72 | 83.1 | 1.03 | 55.59 | ± 1.41 |
| SO201-2-DR87-1^^ | Shirshov | amph | 1 | | | | | | 7/19 | 54.7 | 1.03 | 65.00 | ± 0.60 |
| SO201-2-DR88-2^^ | Shirshov | plag | 1 | | | | | | 14/15 | 98.6 | 1.30 | 68.90 | ± 1.40 |

Ages calculated relative to 28.201 Ma Fish Canyon sanidine standard[73] using the decay constants of ref. [76].
* Abbreviations: grd = groundmass, plag = plagioclase, amph = amphibole
# Expts Nr = number of experiments; atmospheric $^{40}$Ar/$^{36}$Ar = 298.56 ± 0.62[88]
N = number of plateau steps/number of total incremental heating steps
^^Ages from ref. [32] but the original ages have been recalculated to a TCR-2 age of 28.52 Ma with decay constants of ref. [76].

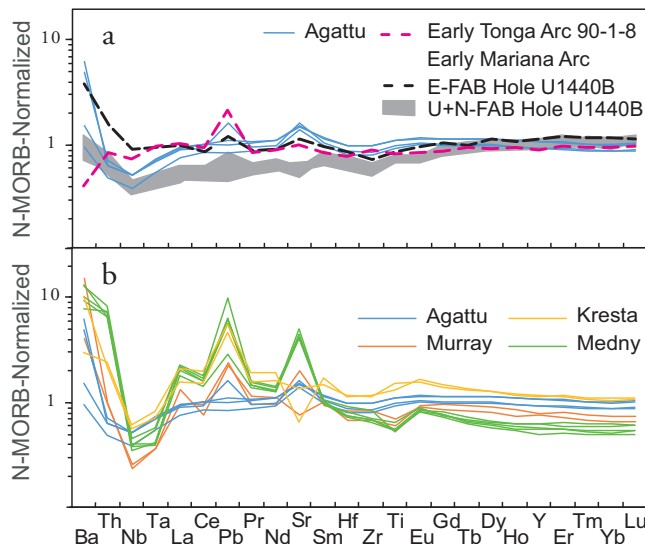

**Fig. 2 | Incompatible-element abundances of oldest Aleutian Arc magmas show east to west spatial variations. a** Agattu forearc basalts with the oldest ages have similar MORB-like trace element compositions (horizontal line at normalized abundance of 1) to volcanic glasses from Mariana enriched (E) forearc basalts (FABs) from IODP Hole U1440B[25] and Tonga forearc[26] tholeiitic basalts. Mariana normal (N) and overlying upper (U) FABs from IODP Hole U1440B are also shown[25]. **b** Murray, Kresta and Medny basalts generally have higher abundances of Ba, Th, LREE, Pb and Sr (except Sr and Ba for two samples having undergone plagioclase fractionation) compared to Agattu FABs and N-MORB, consistent with greater amounts of a sediment component in them. Incompatible-element diagrams normalized to N-MORB after[92].

depth of 100 km above which arc melts are typically generated, providing a minimum age for subduction initiation of 56.1-56.6 Ma.

Specific geochemical lava types are commonly associated with arc initiation. The most detailed studies of subduction initiation have been conducted on the Izu-Bonin-Mariana Arc (IBM), where subduction is believed to have been initiated by 52.5 Ma with volcanism beginning at ~ 52 Ma[18,22–24]. The oldest lavas associated with IBM subduction initiation are MORB-like forearc basalts (FABs) and dolerites, followed by boninites ( ~ 51.3-45 Ma) and then tholeiitic and calc-alkaline basalts at ~45 Ma (Fig. 4;[22–24]). On a multi-element diagram (Fig. 2a), glass analyses of normal (N-), overlying upper (U-), and enriched (E-) FABs from Hole U1440B in the Izu forearc[25] and one of the oldest forearc tholeiitic basalts from the Tonga forearc[26] have overlapping incompatible element abundances with the tholeiitic Agattu forearc rocks. The N + U-FABs, however, generally have lower abundances of elements with intermediate incompatibility (La to Eu) than the Mariana E-FAB and the oldest Tonga Agattu forearc rocks. In contrast to the Agattu and Murray rocks that show only minor arc influence, Kresta and Medny lavas show more pronounced enrichment, especially in fluid-mobile elements. As is illustrated on the Pb versus Nd isotope diagram (Fig. 3c), the good negative correlation can be explained by the addition of ~ 2% average Aleutian sediment to the depleted asthenospheric mantle. Since the eastern locations (Agattu and Murray) and the western locations (Kresta and Medny) both cover the first 10 Ma of Aleutian volcanism, the stronger sediment signature in the western lavas—those erupted closer to the accreted Olyutorsky Arc (see below)—appears to be governed primarily by spatial rather than temporal factors, in contrast to the IBM Arc (Fig. 4). One possibility is that larger amounts of sediments were subducted in the west, where westernmost Aleutian volcanism lies adjacent to the Olyutorsky rearrarc. Alternatively, in the present-day Aleutian Arc, the strongest sediment-derived signatures—comparable to those in the oldest Medny and Kresta lavas—occur in the eastern segment, where subduction is near-orthogonal. In contrast, the western Aleutians, characterized by highly oblique subduction, display a weaker sediment signature similar to that of the oldest Agattu and Murray lavas. This along-arc pattern likely reflects a less compressive regime and a longer slab transport path to the sub-arc source in the present-day westernmost Aleutians[27]. In any case, the pronounced sediment signature in the Medny and Kresta lavas is inconsistent with highly oblique subduction at SZI.

Our ages confirm an earlier initiation age for Aleutian subduction than proposed by previous studies, with three ages of 55.49 ± 0.72 Ma, 55.59 ± 1.41 Ma (Agattu) and 55.06 ± 0.19 Ma (Medny) within error. These ages, however, only provide a minimum age for subduction initiation. Assuming a likely maximum convergence velocity of 10–20 cm/a at subduction initiation[21], with 20 cm/a being consistent with the kinematics of our plate tectonic model below, it would have taken 0.5–1.0 Ma for the subducting lithosphere to reach a typical

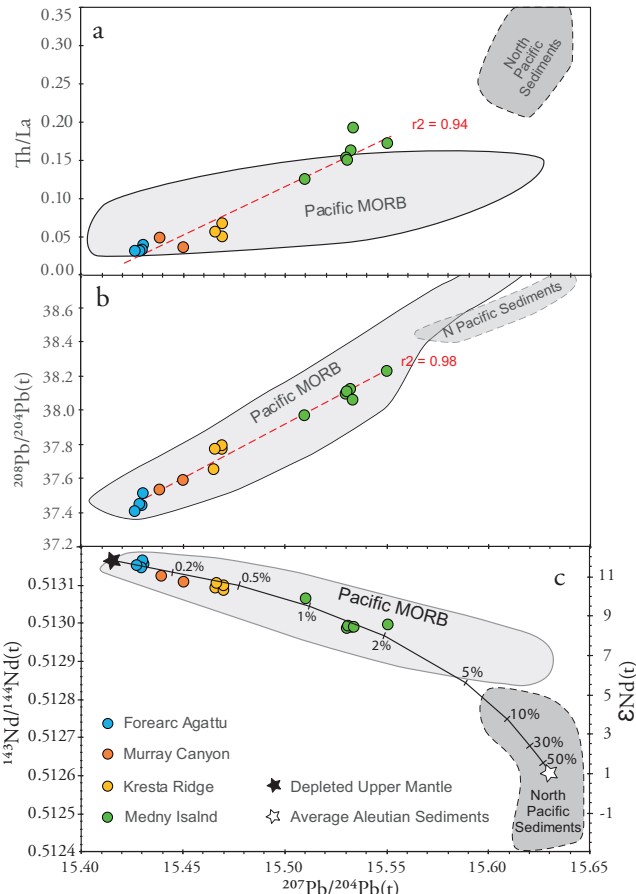

**Fig. 3 | Strong isotope and trace-element correlations of oldest Aleutian magmas demonstrate derivation from the same arc system and mixing between depleted mantle and sediment components.** Initial $^{207}Pb/^{204}Pb$ isotope ratios show good to excellent correlations with (**a**) Th/La trace element ratios, (**b**) initial $^{208}Pb/^{204}Pb$ isotope ratios and c) initial $^{143}Nd/^{144}Nd$ isotope ratios. The correlations point to mixing between a depleted (upper mantle) component with very low Th/La and unradiogenic Pb but radiogenic Nd isotope ratios and an enriched (marine sediment) component with high Th/La and radiogenic Pb and unradiogenic Nd isotope ratios. Correlations of Pb and Nd isotope ratios can be explained by the addition of ~2% average Aleutian sediments to the depleted upper mantle, assuming 85% of the sediment Pb has been removed by fluids under the forearc. Depleted upper mantle end member has 0.023 ppm Pb, 0.71 ppm Nd (from ref. 93), $^{207}Pb/^{204}Pb(t)$ of 15.42 and $^{143}Nd/^{144}Nd(t)$ of 0.51316, assumed isotopic composition based on most depleted Agattu samples. Aleutian sediment endmember from DSDP Site 183 (south of Aleutian arc) has 1.9 ppm Pb, 19.1 ppm Nd, $^{207}Pb/^{204}Pb(t)$ of 15.63 and $^{143}Nd/^{144}Nd(t)$ of 0.51261 (from ref. 94). Tick marks on mixing curve indicate percent of sediment component added to upper mantle. Field for North (N) Pacific Sediments based on data from ref. 94 and for Pacific MORB based on data from PetDb. Errors are within the symbol sizes.

In contrast to the IBM initiation sequence, no boninites have been found in the Aleutians thus far[5] or in our samples. Considering the MORB-like composition of Agattu forearc and IBM FAB samples, boninites might be expected to succeed the FABs. All the Medny (55-46 Ma), Kresta Ridge (>48 Ma), and Murray Canyon (up to 48 Ma) samples, however, have tholeiitic to calc-alkaline affinities, similar to the Tonga Arc, where no boninites were found during the first 10 Myr of subduction initiation[26]. This suggests that the processes at the initiation of the Aleutian and Tonga Arcs were different from those at the IBM. Due to a lack of mantle corner-flow in the wedge at IBM shortly after SZI, hydrous fluids/melts from the slab presumably caused progressive melting of the IBM mantle wedge to produce

boninites[22,25]. It took ~8 Myr for return flow to develop at the IBM Arc, producing tholeiitic and calc-alkaline lavas beginning at ~45 Ma (Fig. 4d). The presence of tholeiitic and calc-alkaline basalts at Aleutian and Tonga SZI suggests that corner-flow was established shortly after arc initiation in their mantle wedges. This is likely to reflect more compressive (induced) conditions during their SZI compared to the IBM.

## Model for origin of Aleutian subduction and backarc basin

Two end-member models have been proposed for Aleutian SZI: (1) Beringian arc splitting with backarc spreading, and (2) plate capture[14]. Although backarc spreading explains the formation of many marginal basins, multiple lines of evidence are inconsistent with this mechanism for the Aleutians. Arc splitting driven by slab rollback would require pre-existing arc or continental crust beneath the Aleutian Arc; however, despite extensive submarine sampling and studies of basal sequences on the Komandorsky and Attu Islands, no such material has been identified ([2,4,5,17] and this study). In addition, the backarc spreading model predicts westward propagation of the Aleutian Arc from the Alaska Peninsula as the Bowers Ridge migrated westward[14], which is also implied by most published Aleutian SZI models[9,11–14]. This prediction conflicts with our age data (Fig. 1), which indicates that the western Aleutians (≥56 Ma) appear to be older than the eastern Aleutians (≥39 Ma), as well as with the geochemistry of the oldest Aleutian lavas which are consistent with arc initiation. Furthermore, backarc spreading would produce a passive Beringian margin, whereas seismic reflection and gravity data reveal a 4–10 km thick sedimentary sequence subparallel to the margin, interpreted as a sediment-filled trench related to an abandoned Paleogene subduction zone and consistent with arc volcanism along the Beringian margin until ~50 Ma[9,14,28]. Together, these observations argue against Paleogene backarc spreading as the mechanism for Aleutian Basin formation.

We therefore favor the plate-capture model. The timing of Olyutorsky Arc collision with the Kamchatka margin and subduction of the Pacific–Izanagi Ridge at ≥57–56 Ma closely coincides with Aleutian SZI, suggesting a causal link[9–14,29]. In contrast to models invoking westward propagation of subduction from the Alaska Peninsula, the age distribution implies that subduction either initiated in the western Aleutians or began synchronously along much of the arc, with subsequent modification by subduction erosion or incomplete sampling in the east.

The easiest tectonic setting in which to initiate a new arc is at a transform fault and/or fracture zone, which juxtapose crust of different ages[11,30]. It has been hypothesized that the northern and southern Olyutorsky Arc segments were separated by a transform fault, which allowed the southern part, closer to Kamchatka, to obduct earlier (~55-50 Ma) than the northern part (~50-45 Ma)[11]. The fault offsetting the Olyutorsky Arc may have also resulted from the original collision of the Olyutorsky arc with South Kamchatka. The transform fault/fracture zone or newly generated fault would have served as a likely place for Aleutian Arc initiation to the east of the colliding Olyutorsky Arc at ≥56 Ma, induced as a result of the plate no longer being able to move westwards due to arc collision and initial Izanagi-Pacific ridge subduction (Fig. 5 and Supplementary Fig. 3). Formation along a transform fault/fracture zone, together with ~40° counterclockwise rotation of the Shirshov Ridge, is also consistent with the magnetic lineations on the Aleutian Basin seafloor being oriented roughly N-S or roughly perpendicular to the Aleutian arc[11]. It is also consistent with paleomagnetic data pointing to formation of the western Aleutians further south or at similar latitudes to the present day (see summary in ref. 14). Although most models call for a polarity reversal[10,11,13] or jump in subduction from the Beringian to the Aleutians[9,12,14], we do not see the necessity for either. It is unclear when the proto-Kamchatka subduction began, but we place it at 53-52 Ma together with subduction re-

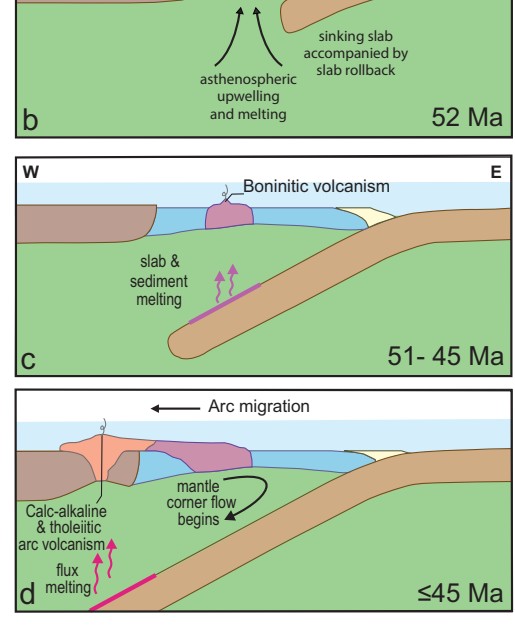

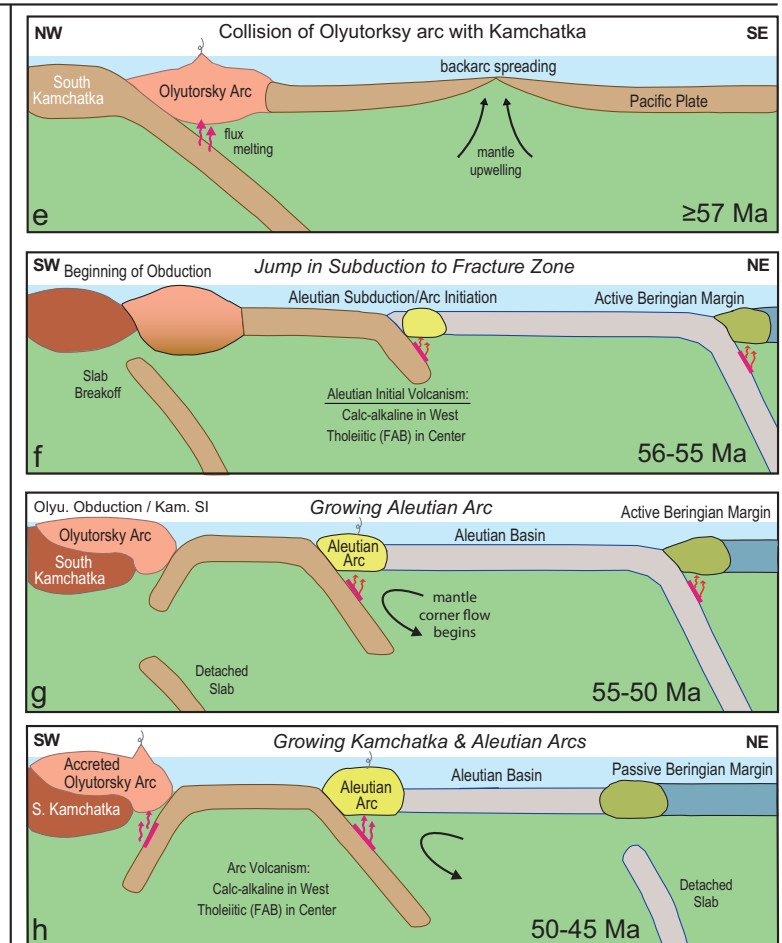

**Fig. 4 | Comparison of Izu-Bonin-Mariana (IBM) with Aleutian and Kamchatka subduction/arc initiation illustrates differences in subduction initiation processes and chemistries of oldest rocks.** Izu-Bonin-Mariana (IBM) subduction/arc initiation (a-d; figures created by the authors, based on models presented in refs. 22–24): E-W profiles: (**a**) 53 Ma: Proto-Philippine and Pacific Plate are separated by a transform fault. Manus plume may have caused the uplift of proto-Philippine Plate. **b** 52 Ma: Possible uplift of the proto-Philippine Plate coupled with sinking of the old Pacific Plate triggered subduction initiation at ≥52.5 Ma, resulting in upwelling of upper mantle and decompression melting to form forearc basalts (FAB) with minor subduction signature, generating new forearc crust. **c** 51-45 Ma: Boninites, andesites, and their differentiates form by flux remelting of residual mantle from which the FABs were derived. **d** ≤45 Ma: Corner-flow replenishes the mantle wedge. Flux melting generates arc tholeiites and calc-alkaline volcanic rocks. **Aleutian/Kamchatka subduction/arc initiation (e–h):** NW-SE profile:

**e** ≥57 Ma: Collision of Olyutorsky Arc with southern Kamchatka. Backarc spreading takes place behind the Olyutorsky Arc. SW-NE profiles: (**f**) 56-55 Ma: Subduction shifts to backarc fracture zone, initiating Aleutian subduction and arc volcanism with the eruption of tholeiitic FAB-like lavas in the east and calc-alkaline lavas in the west, suggesting more oblique subduction and possible extension in the east. Possible Olyutorsky slab breakoff. Active subduction along the Beringian margin. **g** 55-50 Ma: Obduction of Olyutorsky Arc onto southern Kamchatka. Subduction beneath Kamchatka begins at ~53 Ma. Aleutian Arc continues to grow. Subduction continues beneath the Beringian margin. **h** 50-45 Ma: Obduction of Olyutorsky Arc shifts to North Kamchatka (not shown on profile). Kamchatka and Aleutian Arcs continue to grow with tholeiitic volcanism in the east with enhanced arc signature (oldest Murray Canyon lavas) and calc-alkaline in the west (Medny lavas). The Beringian margin becomes passive. Abbreviations: Olyu. = Olyutorsky, Kam. SI = Kamchatka Subduction Initiation.

initiating after Izanagi-Pacific Ridge subduction and with IBM and Tonga SZI (Figs. 4, 5 and Supplementary Fig. 3). Thus, as a result of Olyutorsky Arc collision (probably in its early stages of obduction, not recorded in subaerial outcrops), subduction began at the transform fault/fracture zone separating the northern from the southern part of the Olyutorsky Arc offshore Kamchatka at ≥56 Ma. Transference of subduction from the Olytorsky to the Aleutian Arc does not represent a polarity reversal but rather a jump and clockwise rotation in subduction direction. Polarity reversal did not occur until the Proto-Kamchatka SZI. Finally, considering an overlap in volcanism between the Aleutian and Beringian margins, the Beringian subduction appears to have been slowly eclipsed by Aleutian subduction rather than representing a subduction jump. A re-evaluation of the timing and duration of Beringian Margin volcanism with modern geochronologic methods is warranted.

The Shirshov and Bowers Ridges form prominent bathymetric highs on the Aleutian Basin Seafloor (Fig. 1A). In early models, these ridges were considered to be either a lateral (but folded) extension of the Cretaceous Olyutorsky Arc or formed by Alaskan continental extrusion during the early Eocene (see summary in ref. 9). Therefore, we need to consider their possible relationship to Aleutian SZI. The Shirshov Ridge is connected to the accreted Olyutorsky Arc in the north at the Olyutorsky Peninsula. It consists of late Cretaceous arc-type volcanic rocks with Ar/Ar ages of 66.5 ± 0.6 Ma and 70.5 ± 1.4 Ma (Table 1;[31]) from the central western Shirshov Ridge. Gabbroic sequences from northern Shirshov Ridge yielded zircon ages of 64-80 Ma (n = 21) with a mean age of 72 ± 1.4 Ma (n = 18)[32], which overlap with the arc volcanic rocks and thus may also be derived from magmatism forming the ridge. Taken together, these ages suggest Shirshov belonged to an older arc than the Aleutians, most likely a piece of

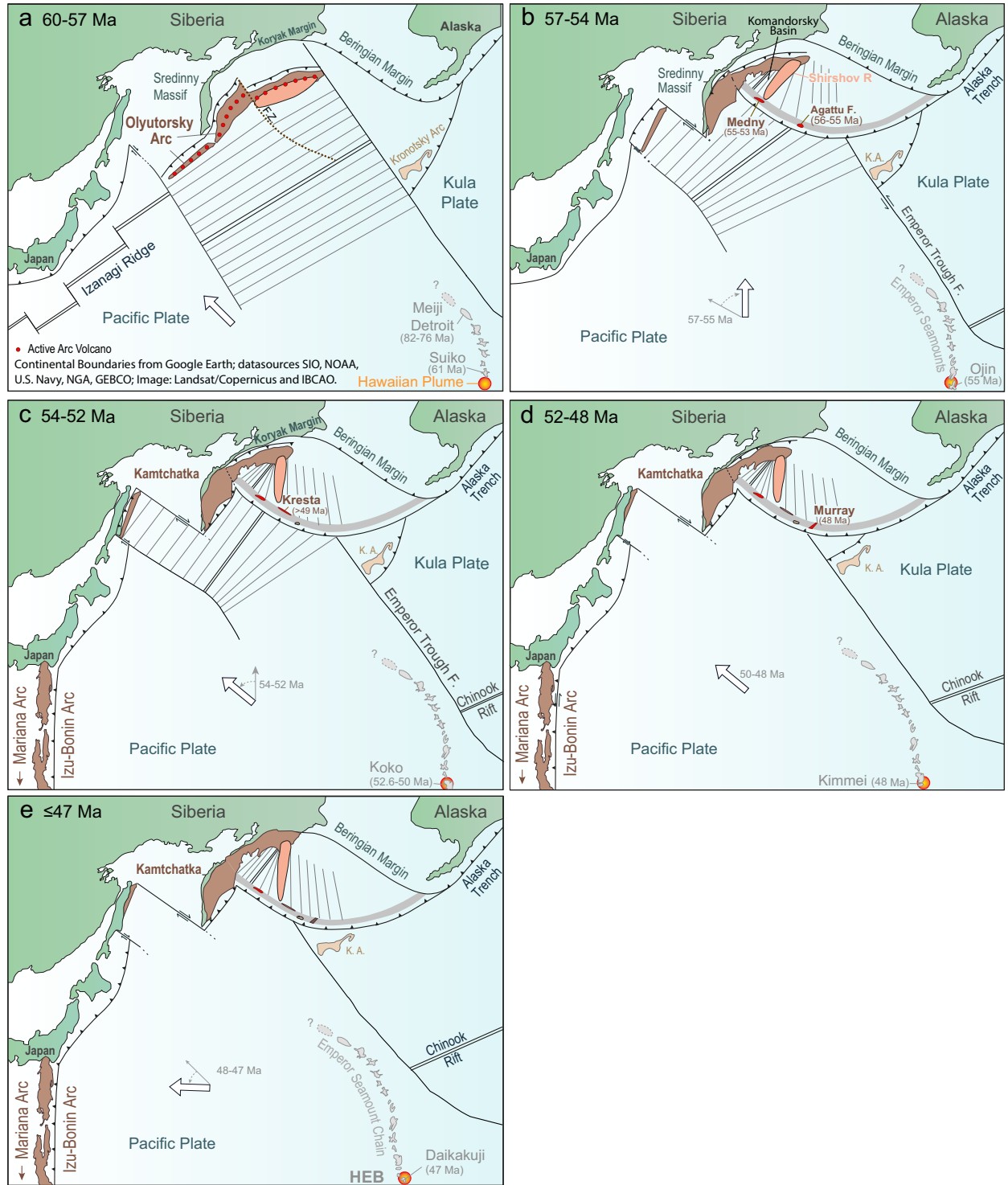

the northern Olyutorsky Arc, with which it is connected[11]. Evidence for the opening of the proto-Komandorsky Basin in the early Eocene comes from exotic rock fragments (jasper, quartzite, slate, schist, gabbro and granite) and minerals (e.g., metamorphic quartz and garnet) in sandstones in the early Eocene Komandorsky Series on Medny Island, which have not been found elsewhere on Medny or on the neighboring Bering Island[17]. The sandstones are deposited in a marine fan with a source area that was to the north of Medny Island. The rocks are similar to those from the Shirshov Ridge and Eastern Kamtchatka,

belonging to the Olyutorsky Arc[17]. Although they were interpreted as having been derived from the Shirshov Ridge as Medny possibly moved ~ 200 km westwards along the Bering-Kresta right-lateral strike-slip fault in the early Eocene[17], we believe that it is more likely that they were derived from the ridge as it rotated counterclockwise away from the forearc portion of the northern Olyutorsky Arc triggered by Aleutian Arc formation (Fig. 5). The rotation of the Shirshov Ridge was accommodated by subduction at the Beringian margin[11], which lasted until 50 Ma[28] and coincides with the initiation of obduction of the

**Fig. 5 | Schematic model for Aleutian and Kamchatka Subduction/Arc Initiation and evolution during the late Paleocene – early Eocene Pacific plate reorganization (60 to ≤ 47 Ma). a** 60-57 Ma: Olyutorsky Arc, with SE dipping subduction beneath it, begins to collide with the southern Kamchatka margin. The arc offshore of Kamchatka-Koryak margin is offset by a right-lateral transform fault leading into a fracture zone (F.Z.) cutting late Cretaceous-Paleocene (< 85 Ma) crust formed by backarc spreading. Seafloor spreading anomalies are oriented NE-SW. NW dipping subduction of the Kula Plate occurs beneath the Kronotsky Arc. **b** 57-54 Ma: Obduction of Olyutorsky Arc onto southern-central Kamchatka begins. Aleutian subduction initiates along the backarc fracture zone at ≥ 56 Ma. Aleutian Arc volcanism begins in the Agattu forearc and on Medny Island. Plate motion changes from NW to N between 57-55 Ma. Aleutian SZI triggers the splitting of the northern Olyutorsky arc segment offshore North Kamchatka and Koryak margin, which begins to rotate eastwards (counterclockwise), opening the proto-Komandorsky Basin. The seafloor anomalies formed by Olyutorsky backarc spreading also begin to rotate counterclockwise. **c** 54-52 Ma: Shirshov Ridge continues to rotate counterclockwise, opening the proto-Komandorsky Basin,

prolonging subduction at the Beringian margin. Plate motion shifts from N back to NW between 54 and 52 Ma. Due to the subduction of the Izanagi-Pacific Spreading Center, Pacific Plate subduction begins again along the Japanese margin and further south. NE-oriented proto-Kamchatka subduction begins between 53-52 Ma beneath the Olyutorsky arc segment, which continues to obduct onto southern Kamchatka until ~ 50 Ma. Izu-Bonin-Mariana (and Tonga) subduction initiates at ≥ 52.5 Ma. **d** 52-48 Ma: Pacific Plate subduction continues to the NW. The northern Olyutorsky Arc begins to obduct onto the northern Kamchatka-Koryak margin. Proto-Komandorsky Basin stops opening, and Beringian margin subduction ceases at ~ 50 Ma. Polarity reversal of Kronotsky Arc at 50-47 Ma[39]. Hawaiian-Emperor Bend (HEB) begins to form. **e** ≤ 47 Ma: Plate motion shifts to the W between 48-47 Ma due to enhanced slab pull along the western Pacific margin, with a possible contribution from Kronotsky Arc polarity reversal[39]. Bend in the Hawaiian-Emperor seamount chain completed. Abbreviations: Emperor Trough Fault (F.). Continental outlines in all panels are from Google Earth; datasources: SIO, NOAA, U.S. Navy, NGA, GEBCO; Image: Landsat/Copernicus and IBCAO, as indicated in panel (**a**).

---

northern Olyutorsky Arc to the Kamchatka-Koryak margin[11]. Therefore, we postulate opening of the proto-Komandorsky Basin between ≥ 56 and ~ 50 Ma, consistent with ref. 11.

As is the case with the Shirshov Ridge, few samples have been recovered from the Bowers Ridge. The existing data on these samples show that it was an active volcanic arc between 34 and 22 Ma, as the Bowers backarc was opening, with a trench on its NE side[33,34]. Thus far, there is no evidence for volcanism before the Oligocene as proposed in many earlier models. More recently, it has been proposed that the Bowers Ridge was originally a fragment of the Aleutian Arc knocked out through a collision with the Kronotsky Arc, causing a polarity reversal of a part of the Aleutian Arc[11,35]. This model, however, is not consistent with the age data. Numerous Ar/Ar ages from Murray and Attu canyons and Kresta Ridge show that this part of the arc was active almost continuously from 49 to at least 14 Ma[4,5], covering the time during which Bowers was active[33,34]. Therefore, it is unlikely that the Kronotsky Arc collision caused a gap in the Aleutian Arc, but this does not rule out that a part of the Aleutian reararc split off upon collision, similar to the Shirshov Ridge splitting off from the Olyutorsky Arc upon Aleutian SZI. The width of the Aleutian Arc massif decreases towards the west from the Bowers-Aleutian junction, consistent with a part of the arc having been removed. Right-lateral movement along the Alpha and/or Bering-Kresta faults, combined with backarc spreading related to westward-dipping subduction in the late Eocene[33] could explain the formation of the Bowers Ridge.

## Paleogene Pacific plate reorganization

We now place Aleutian subduction initiation in the context of the Paleocene-Eocene plate tectonic reorganization in the Pacific Basin. In the northern Pacific Basin, the NE-SW-oriented Olyutorsky Arc was active from ~ 85 Ma until collision with Kamchatka between 60-55 Ma, followed by obduction of its central segment onto South Kamchatka at ~ 55-50 Ma and of its northern segment at ~ 50-45 Ma[11]. Contemporaneous volcanic activity also took place in the NE-SW-oriented Chukotka and Alaska ranges (77-56 Ma) and the Alaska forearc (63-56 Ma) (Fig. 1a)[9,12,28,36,37]. At ~ 56 Ma, volcanism ceased in these areas and shifted to the NW-SE-oriented western Aleutians, Beringian margin and North American margin. Some of the marginal terranes, such as the Sanak-Baranof belt, however, may have formed further south and translated northwards by dextral shear along the northern Cordilleran margin[36].

Since none of the published reconstructions proposes Aleutian SZI at ≥ 56 Ma, we develop a plate model that incorporates our Aleutian SZI age. The model evaluates the role of the Aleutian SZI in the ~ 10 Myr (57-47 Ma) major plate tectonic re-organization ending with the Hawaiian-Emperor Bend, which is increasingly believed to be related to both a plate motion and mantle flow change (plume dynamics) from 50

to 47 Ma[7,38–41]. Furthermore, we incorporate the Olyutorsky and the Kronotsky Arcs into our model, because studies have shown that multiple North Pacific intra-oceanic subduction zones in the late Cretaceous to early Paleogene produce better results than single subduction zones[10,11], e.g., higher correlations to observed residual topography in NE Asia[42], and there is geological evidence for both subduction zones from arcs accreted to Kamchatka[11,20].

We modified the GPlates global plate motion model[43], holding Antarctica fixed, to incorporate Olyutorsky, Aleutian, Kamchatka and Kronotsky arc evolution and accretion, as well as changes in the Pacific Plate boundaries in the western Pacific Basin. Most notably, these changes include the interruption of subduction between 56 and 53 Ma after subduction of the Pacific-Izanagi Ridge, and the ≥ 52.5 Ma onset of IBM subduction along the Philippine Sea Plate (Fig. 6 and Supplementary Fig. 3). Between > 60 and 57 Ma, modeled Pacific and Kula Plate motions were directed toward the NW (~ 300°). During this interval, collision of the NE-SW-oriented Olyutorsky Arc with SE-facing subduction beneath it commenced, while NW-facing subduction was active beneath the Kronotsky and Alaska Arcs. Our model shows that between 57 and 55 Ma, primarily between 56 and 55 Ma, Pacific Plate motion shifted clockwise to ~ 350° north, correlating with our age for Aleutian SZI. A clockwise rotation in plate motion of ~ 40° is also observed in global relative plate motion circuits if the North American Plate is fixed. Since the Pacific Plate was moving to the NW before 56 Ma, the trigger for this plate motion change is likely to have been reduced subduction towards the west due to cessation/interruption of subduction in the western Pacific. The collision of the Olyutorsky Arc with the Kamchatka margin between 60 and 56 Ma[11] and the subduction of ~ 2700 km of the Izanagi-Pacific spreading center between 60 and 56 Ma[29,44] appears to have blocked the westward component of plate motion, resulting in a shift in subduction towards the N and Aleutian SZI (Figs. 5, 6). Seismic tomography under eastern Asia (between 500 and 1000 km depths) reveal a pronounced discontinuity, consistent with slab detachment and a brief lull in subduction between 60 and 50 Ma[45]. Slab detachment would have diminished slab pull to the west, which would have contributed further to a major change in plate-motion rate and direction. Aleutian SZI and continued Pacific Plate subduction beneath the Kronotsky Arc and Alaska would have contributed to northward slab pull. A shift to northward plate motion can also explain why magmatism largely ceased along the NE-SW oriented portion of the Alaska subduction zone at c. 56 Ma and shifted to the NW-SE-oriented western Aleutians and eastern Alaska-British Columbia margin at that time[9,12,36].

Re-initiation of subduction outboard Japan (where the Pacific-Izanagi Ridge subducted) and beneath southern Kamchatka (the obducting Olyutorsky Arc) at ~ 53 Ma and SZI of IBM at ≥ 52.5 Ma and Tonga at ~ 52 Ma[22,26,29] are consistent with a counterclockwise rotation

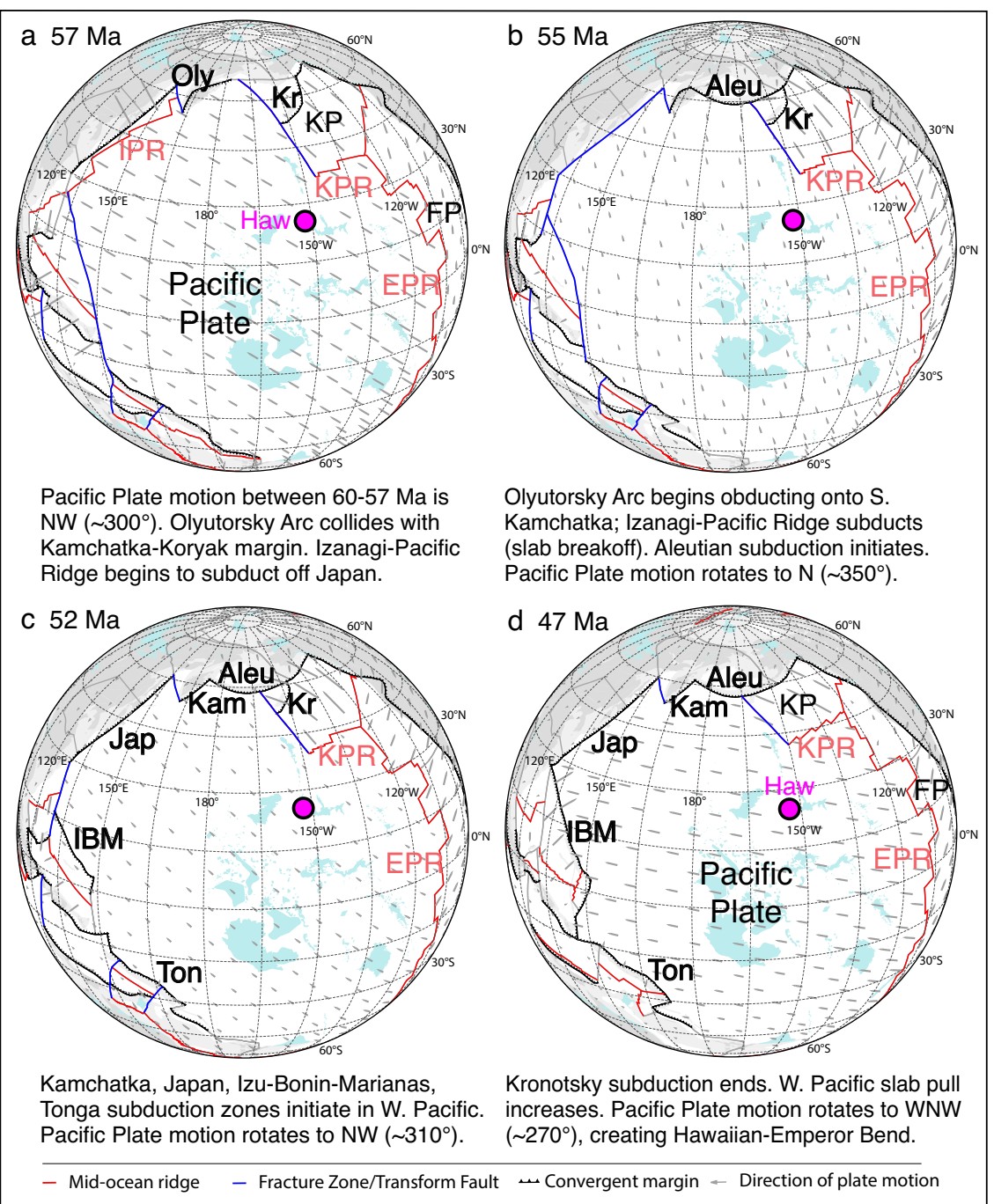

**Fig. 6 | Plate tectonic reconstructions define three sets of Pacific Plate motion changes between 57-47 Ma.** GPlates-based reconstructions in an Antarctica fixed reference frame[43] at (**a**) 57 Ma, (**b**) 55 Ma, (**c**) 52 Ma, (**d**) 47 Ma. The plate motion changes are related to Aleutian (57-55 Ma) and Kamchatka-Japan-IBM-Tonga (54-52 Ma) subduction/arc initiation, and formation of the Hawaiian-Emperor Bend (50-47 Ma) due to increased slab pull in the western Pacific[45], and slow-down of southward drift of the upper Hawaiian plume (50-47 Ma;[41]). After 47 Ma, plate motion has remained relatively constant until the present-day. See Supplementary

Fig. 3 for additional projections from 60-40 Ma. Abbreviations: Subduction zones/arcs in black: Oly = Olyutorsky, Kr = Kronotsky, Aleu = Aleutian, IBM = Izu-Bonin-Marianas, Ton = Tonga; Mid-ocean ridges in red: IPR = Izanagi-Pacific Ridge, KPR = Kula-Pacific Ridge, EPR = East Pacific Ridge; Plate names in black: KP = Kula Plate, FP = Farallon Plate; Hotspots: HAW = Hawaiian; and large faults in blue. Paleogeographic reconstruction data modified from[95,96] licensed under CC BY 4.0 (https://creativecommons.org/licenses/by/4.0/).

in plate motion from NNW to NW (~310°) between 54 and 52 Ma (Figs. 5, 6). This plate motion change can also explain the counter-clockwise rotation of the Kula spreading center (shown by the magnetic anomalies just south of the Aleutian Trench[21]; Fig. 1b) from E-W to a NE-SW orientation, considering that the Kula Ridge was located 1500-2000 km south of the Aleutian subduction zone between 53.9 and 52.5 Ma, when the rotation in magnetic anomalies began using

more recent ages for the magnetic anomalies[46]. The contemporaneous emplacement of the Siletzia Large Igneous Province (LIP) onto the North American margin would have affected subduction of the Farallon Plate[47]. Although older ages suggested inception of Siletzia volcanism at ~56 Ma, recent U/Pb zircon ages and calcareous nanoplankton ages of interbedded marine mudstones constrain the formation of Siletzia to ≥53.5–48.2 Ma[47–49]. The accretion of Siletzia

undoubtedly affected the change in spreading direction between the Farallon and Pacific Plates between chrons 24-21 (~53-48 Ma)[50] and a major change in spreading direction between the Kula and Pacific plates at chron 24 (~53 Ma)[45], which would also have influenced Pacific Plate motion at 54-52 Ma. The accretion could also possibly have triggered hinterland orogenic collapse and formation of the ~3000 km-long Challis-Kamloops belt of magmatism and extensional core complexes (53-45 Ma;[51]) extending from central British Columbia to southern Idaho (Fig. 1a). Finally, cessation of spreading in the Tasman and Coral Seas east of Australia at ~52 Ma[52], westerly migration of Wharton Basin spreading center increasing northward slab pull of Australia, and a change in Australian plate motion NW to N at ~52 Ma also influenced Pacific plate motion and IBM and Tonga SZI[29].

It has been proposed that the India-Tibet collision could have been linked to subduction initiation in the western Pacific[24]; however, collision age estimates range from early Paleocene (62.7-61 Ma;[53]) to Oligocene (~34 Ma;[54]). Increasing evidence suggests that the original collision was between Greater India and an inter-oceanic island arc rather than the Asian continent[54]. It is unlikely that the collision of India with an island arc would have had a major impact on subduction initiation in the western Pacific in the early Eocene.

In our model, Pacific plate motion remains relatively constant from 52 to 48 Ma and then changes to nearly due west between 48 and 47 Ma (Fig. 6 and Supplementary Fig. 3). Although some of the plate motion change may result from Kronotsky slab breakoff and polarity reversal between 50 and 47 Ma[39], the main cause is likely to be increased slab pull from IBM, Tonga, Kamchatka and Japanese subduction in combination with other circum-Pacific events[29,45]. The change to westward plate motion, combined with a slowing of the southward drift of the Hawaiian hotspot[41,55], generated the Hawaiian-Emperor Bend[7,38-40]. Recently, Zhang & Hu[40] proposed that the Hawaiian plume drifted southwestwards as the plate moved westwards. Assuming that the westward component of plume motion balanced westward plate motion, this would explain why the Emperor seamounts form a nearly N-S chain of seamounts on the seafloor.

In conclusion, widespread subduction of the Izanagi–Pacific ridge, accompanied by slab breakoff and slab-window formation, together with collision of the Olyutorsky Arc with the Kamchatka–Koryak margin, leading to obduction and additional slab breakoff, triggered a cascade of tectonic and magmatic processes across the Pacific region. These included multiple SZI events (Aleutian, Kamchatka, Japanese, IBM and Tonga) and collision of the Siletzia LIP with the North American margin. Acting in concert along the circum-Pacific, these closely linked processes modified seafloor plate motions and drove a protracted (10 Myr) plate reorganization in the Pacific region during the late Paleocene to early Eocene (≥57–47 Ma), culminating in the formation of the Hawaiian–Emperor Bend, which was further influenced by motion of the Hawaiian plume[41,55].

## Discussion

We will now briefly discuss the possible climatic impact of Olyutorsky Arc obduction and Aleutian SZI, including other Pacific subduction initiation events. The Cenozoic hothouse (56-47 Ma), overlapping with the plate reorganization, was initiated by a short episode of extreme global warming at the Paleocene-Eocene boundary (56.01 ± 0.05 Ma) that was associated with dramatic negative O and C isotope excursions[56,57]. It lasted for 170 ± 30 kyr from onset to recovery[57], indicated by an increase in sea surface temperatures of 8–10 °C and deep-sea temperatures of 4-5 °C at high latitudes and 4-5 °C in the tropics[58] and is commonly referred to as the Paleocene-Eocene Thermal Maximum (PETM). The origin of this event is highly relevant for understanding the Earth's response to present-day anthropogenic-driven climate change, although its origin remains highly controversial[58-60]. Although some studies have proposed volcanism related to the North Atlantic Igneous Province as being the source of

the C forming the PETM[60,61], a recent study using mercury as a proxy for volcanic activity did not find a Hg anomaly in North Atlantic sediment cores covering the PETM[62].

Our study shows that Aleutian SZI coincided with the onset of the PETM, suggesting that it may have played a role in triggering this event. We envisage two processes that could be responsible for releasing large amounts of C during SZI. Intrusion of melts during the initiation of subduction zone volcanism into carbon-rich volcanic turbidites in the Olyutorsky backarc could have resulted in a sudden release of large amounts of carbon into the ocean and atmosphere due to induced hydrothermal venting[60,61]. The coeval collision of the Olyutorsky Arc with Kamchatka and the onset of the Aleutian arc volcanism with the PETM also revives an alternative explanation for the source of carbon that forced the PETM. Higgins and Schrag[63] proposed that the isolation of a large epicontinental seaway by tectonic uplift associated with volcanism or continental collision, followed by desiccation and bacterial respiration of the aerated organic matter, is another potential mechanism for the rapid release of large amounts of $CO_2$. So far, this hypothesis has been discarded because no suitable tectonic event could be identified. The collision of the Olyutorsky Arc with the Kamchatka continental margin and underthrusting of the Pacific Plate beneath the North American Plate (Aleutian SZI), however, would have uplifted the over-riding plate. Uplift and exposure of an epicontinental seaway and/or hydrothermal activity in the sedimentary basins around the young arc volcanoes could well have injected large amounts of light carbon into the oceans and atmosphere. While there is no geological data yet to corroborate a direct link between Aleutian SZI and the PETM, the striking coincidence in age means that such a connection should be considered. Aleutian SZI may well have also played a key role in explaining the pattern of carbon injection into the Earth system at the onset of the PETM. Finally, our study suggests that SZI should be considered as a potential mechanism for triggering global warming events more generally, such as the Cenozoic hothouse.

## Methods

### Zircon U-Pb dating

Zircon age data are included in Table 1 and Supplementary Table 1, Fig. 1. In-situ zircon U-Pb isotopes were measured using a Thermo Element XR SF-ICP-MS coupled with a Resonetics Resolution 155 type ablation system at ETH Zürich. We used a 19 μm spot size, 4 Hz repetition rate, 2.0 J/cm². energy density (fluence) and 30 s ablation time after five cleaning pulses and 30 s of gas blank acquisition. For U-Pb dating, GJ-1 reference zircon[64,65] was used as a primary reference material, while zircons 91500, Plešovice, AUSZ7-1, AUSZ7-5, GHR-1, and Rak-17 were measured as validation reference materials ([66-71], respectively). The average precision of these reference materials (RM) ranged from 2.0% to 10.5% (2 SE) while the average precision of the samples is 3.8% (further details are in Supplementary Table 1). Data reduction of in-situ dating by LA-ICP-MS was done using IOLITE 4[72] combined with VizualAge[73]. The in-situ dates were not corrected for common Pb contents; however, during data reduction, integration intervals were set to exclude the common Pb contaminated signal intervals, and data were filtered according to their discordance [($^{207}Pb/^{235}U$ Age) - ($^{206}Pb/^{238}U$ Age)]/($^{207}Pb/^{235}U$ Age) < 10%). For final ages, a systematic uncertainty of 1.5% was propagated quadratically, including uncertainties from the not corrected possible common Pb contribution, the uncertainty of the age of the primary Reference Material (RM), and the uncertainty of the applied corrections (downhole fractionation, drift). Th disequilibrium is not corrected as the usual correction is well within the overall uncertainty (Supplementary Table 1).

### $^{40}Ar/^{39}Ar$ dating

Groundmass (180–250 μm) was ultrasonically leached in 3 M HCl for ten minutes, rinsed repeatedly with deionized water, and then hand-

picked under a binocular microscope. Plagioclase was subject to additional leaching in 10% HF for five minutes. The purified separates were irradiated in the cadmium-lined in-core tube at the Oregon State University reactor. The 28.201 Ma Fish Canyon sanidine[74] was used as a neutron fluence monitor for several irradiations containing these samples. $^{40}Ar/^{39}Ar$ analyses were conducted in the WiscAr Laboratory at the University of Wisconsin-Madison. Groundmass aliquots ( ~ 2–7 mg) were placed in a 2 mm diameter well in a copper tray and incrementally heated with a 55 W $CO_2$ laser. The gas released during each heating step was cleaned with two SAES GP50 getters (50 W/ 400 °C) and an ARS cryotrap (at − 125 °C). Isotopic analyses were done using a Nu Instruments Noblesse five-collector mass spectrometer[75]. Plagioclase from sample SO249-DR51-6 was analyzed using an Isotopx NGX-600 mass spectrometer[76]. All of the $^{40}Ar/^{39}Ar$ ages are calculated using the decay constants of ref. 77 and are reported with 2σ analytical uncertainties, including the J uncertainty (Table 2, Supplementary Table 2 and Fig. 2).

### Whole rock geochemistry

Major and selected trace elements were determined on a PanAnalytical MagixPro PW2540 X-ray fluorescence spectrometer at the Institute of Mineralogy and Petrography at the University of Hamburg, Germany (Supplementary Table 3). Reference materials JGb-1, JB-2, JB-3, JA-3, and JG3 were measured as unknowns along with the samples. The major elements lie mostly within 3% and trace elements within 10% of the Govindaraju[78] reference values. High-precision trace element compositions were determined by inductively coupled plasma mass spectrometry (ICP-MS). With the exception of zircon-bearing rocks, samples were decomposed on the hotplate in HF-HNO3-HCl-HClO4 and analyzed in solution by ICP-MS on an AGILENT 7500cs at the Institute of Geosciences of the Christian-Albrechts-University of Kiel[79]. For zircon-bearing rocks, trace element concentrations were obtained two ways: a) two digestion steps were done using HNO3-HF and Parr bomb vessels[79]; or b) via the laser-ablation-ICPMS (LA-ICPMS) measurement of nano-particulate pressed powder tablets, using a 193 nm excimer laser ablation system (GeoLas Pro; Coherent) coupled to an Agilent 7900 ICP-MS[80], where concentrations presented are the average of 3 spots (SO249 DR40-2, 40-2xen). For each batch of samples, replicate analyses gave a relative standard deviation typically less than 3% for all elements reported. BHVO-2 and BIR-1 were measured in every batch of samples measured in solution, and BHVO-2 and KL2-G were measured during LA-ICPMS analyses of nano-particulate pressed powders (Supplementary Table 5).

Radiogenic isotope ratios of Sr, Nd and Pb were determined by thermal ionization mass spectrometry (TIMS) at GEOMAR (Supplementary Table 4). Prior to dissolution, sample chips were leached in 2 M HCl at 70 °C for one hour and thereafter triple rinsed in 18MΩ $H_2O$. NBS987 (Sr) and La Jolla (Nd) were measured 4-5 times each wheel, and the average $^{87}Sr/^{86}Sr$ and $^{143}Nd/^{144}Nd$ were used to obtain a factor to normalize the measured data to the reference values. Thus, the sample data is reported relative to $^{87}Sr/^{86}Sr = 0.710250 \pm 0.000008$ (2 SD, $n = 181$) and $^{143}Nd/^{144}Nd = 0.511850 \pm 0.000006$ (2 SD, $n = 581$). Sr and Nd within run mass bias correction uses $^{86}Sr/^{88}Sr = 0.1194$ and $^{146}Nd/^{144}Nd = 0.7219$. Pb isotope ratios were determined by Pb double-spike (DS)[81]. NBS981-DS values are $^{206}Pb/^{204}Pb = 16.9408 \pm 0.0019$, $^{207}Pb/^{204}Pb = 15.4975 \pm 0.0019$, $^{208}Pb/^{204}Pb = 36.7206 \pm 0.0050$, $^{207}Pb/^{206}Pb = 0.914801 \pm 0.000048$ and $^{208}Pb/^{206}Pb = 2.167858 \pm 0.000097$ (2 SD, $n = 205$). BCR-2 was processed similarly to the samples. Total procedural blanks were typically < 30 pg Pb, < 100 pg Sr and < 50 pg Nd.

### Plate tectonic reconstructions for 60 to 40 Ma

The Müller et al[43]. plate motion model was modified for the North Pacific for 60 to 40 Ma following our interpretations in Figs. 4 and 5,

including capturing the subduction polarity and terrane motions related to the Kronotsky and Olyutorsky arc systems and the Aleutian subduction initiation timing proposed here. The underlying plate motion model, in the Antarctica-fixed reference frame, makes use of the magnetic anomaly and seafloor spreading histories documented in the Seton et al ref. 45. compilation, with the Pacific Plate motion tied to the global plate circuit via South Pacific seafloor spreading relative to Marie Byrd Land (West Antarctica), through East Antarctica, with the plate circuit continuing via Antarctica and Africa. Most relevant to this study are the 60 to 40 Ma rotations which are defined by the magnetic anomaly identifications and rotations of the Pacific Plate as reported in Wright et al[82]. for 61.3 Ma (Chron 27o), 56.15 Ma (Chron 25 m), 53.3 Ma (Chron 24n3o), 47.9 Ma (Chron 21o), 43.8 Ma (Chron 20o), and 40.1 Ma (Chron 18o) using the Gee and Kent[83] magnetic polarity reversal timescale. The absolute plate motions are underpinned by the kinematically optimized mantle reference frame presented in ref. 43, which aims to find a compromise between (1) fitting global hotspot tracks, (2) minimizing (but not eliminating) net lithospheric rotation, and (3) minimizing trench advance. Given the anomalous deflection of the Hawaiian-Emperor seamount chain, likely from a combination of plume-ridge interactions (e.g., Madrigal et al.[84]), deflections from the 'mantle wind' (e.g., Tarduno et al.[85]) and basal plume motion (e.g., Hassan et al.[86]), the pre-43 Ma Hawaiian-Emperor chain history was excluded from this mantle frame optimization process. Although the absolute plate velocity vectors in Fig. 6 are the result of a combination of the relative plate motions and the absolute reference frame, the first-order changes in plate motion direction in the North Pacific arise primarily from the seafloor spreading histories rather than the mantle reference frame.

## Data availability

The age and geochemical data generated in this study have been deposited in the Dryad database under accession code doi:10.5061/ dryad.wpzgmsbwx [https://doi.org/10.5061/dryad.wpzgmsbwx][87] and are provided in the Supplementary Information file. All display items containing data, including means/averages, can be reproduced from raw data.

## Code availability

The revised GPlates model and data files generated in this study have been deposited in the Zenodo database under accession code https:// doi.org/10.5281/zenodo.19360646 [https://zenodo.org/records/ 19360646].

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

## Acknowledgements

Reinhard Werner is thanked for help obtaining funding and carrying out the cruises; S. Jung for providing XRF and D. Garbe-Schönberg for ICP-MS data; S. Hauff, K. Junge and U. Westernströer for analytical support; and Deep Sea Drilling Project (DSDP) and International Ocean Discovery Program (IODP) for sediment and IBM forearc reference samples.

## Author contributions

Writing (KH), Review and Editing (K.H., B.J., M.P., S.Z., D.M., F.H., C.T., T.W.H., G.Y., M.G., C.B., D.S., R.B. and B.B.), Data Interpretation (K.H., B.J., M.P., S.Z., D.M., F.H., C.T., T.W.H., G.Y., M.G., C.B., D.S., R.B. and B.B.), Conceptualization (K.H., B.J., M.P., G.Y. and B.B.), Ship and field sampling (K.H., M.P., F.H., G.Y., T.H. and D.S.), Data generation (B.J., F.H. and M.G.), Modeling (S.Z. and D.M.), Visualization (K.H., B.J., S.Z., D.M., C.T., F.H., M.G. and C.B.), Funding acquisition (K.H., B.J., M.P., G.Y., D.M. and S.Z.).

## Funding

K.H., M.P., and F.H. disclose support for the research of this work from the German Federal Ministry of Research, Technology and Space (BMFTR) [grants KALMAR 03G0640A and Bering 03G0249]; BJ from the National Science Foundation (NSF) [grants EAR-1753492 and OCE-1551657]; G.Y. from NSF [grants OCE-1551640 and EAR 1753518]; S.Z. from the Australian Research Council (ARC) [grant DE210100084]; D.M. and S.Z. from the AuScope National Collaborative Research Infrastructure System (NCRIS) program and GPlates Development. The remaining co-authors declare no relevant funding. Open Access funding enabled and organized by Projekt DEAL.

## Competing interests

The authors declare no competing interests.
