## [Transparent Peer Review file · Nature Communications]

Tectonic and climatic implications of the Aleutian Arc initiation ≥ 56 million years ago

Corresponding Author: Professor Kaj Hoernle

Version 0:

Reviewer comments:

Reviewer #1

(Remarks to the Author)

Hoernle and others present new geochronology and geochemistry from dredged arc rocks in the western Aleutians to provide new constraints on the initiation of Aleutian arc volcanism and subduction. Their findings (initiation >56 Ma) are consistent with previously proposed models linking Aleutian subduction with cessation of Alaska Range and Siberian/Koryak volcanism. Overall, the paper is well-written and its intended message is clear/consistent. The study provides novel and important data/results from a remote geologic setting. Moreover, as highlighted by the authors, the geodynamic initiation and history of Aleutian arc is a critical piece to regional Pacific plate reconstructions, which is well-supported.

New results convincingly demonstrate that western Aleutian volcanism began by 56 Ma, resolving any previous uncertainties between older initiation (56-50 Ma) and younger initiation based on more reliable direct arc dating (<50 Ma). The authors interpret (or propose) a causal relationship between coeval Pacific plate reorganizations with warming of the PETM. As I understand it, their argument is that enhanced volcanism associated with these events tipped the scales of climate response against a backdrop of already enhanced volcanism associated with the North Atlantic Igneous Province (NAIP). Although I have no objections to the possibility of these mechanisms, my main concern is that it is based primarily on correlation. Because plate reorg in and of itself is not an actual mechanism, the authors propose that this reorg results in increased global volcanism. This is perhaps the main area of concern in the paper, in its current form. To make a more convincing argument, it would help to show that this plate reorg results in a marked increase in volcanism (or emitted CO₂) at that time. For example, could they total the length of Pacific subduction zone interfaces before and after this plate reorg to show an abrupt change (increase) in volcanism. If not that, then something tangible to better connect a link between how this plate reorg results in substantial increases in CO₂. Moreover, it would also help to clarify if (and why) there's a net gain in volcanism between the loss of the Olyutorsky Arc and the initiation of the western Aleutian Arc (lines 381-382: "it provides an additional large-scale potential carbon source in the North Pacific for this global event").

Lines 39 to 41: Something is off (typo?) in these two sentences. Important details are placed into a string of hard-to-follow parentheses. I suggest just placing this into a simpler sentence format.

Line 74-75: Authors state: "To date, there is no viable explanation for the cause of the PETM." Perhaps there is no uniform consensus on the cause(s) for the PETM but multiple 'viable' mechanisms have been proposed, including those discussed later in this paper.

Line 76: Paragraph begins with "To address these questions", however no questions were presented in the previous paragraph. I suggest either posing some questions or clarifying what is being addressed (and how) by presenting new geochronology and whole rock data (e.g., uncertainties, controversies, gaps in knowledge, etc.).

The link between Aleutian Arc initiation and the PETM is alluded to multiple times early in the manuscript (abstract, lines 70-72, lines 90-91) before the reader gets to the actual proposed causal mechanism (increased C-rich volcanism). Consider at least a brief mention of what the proposed mechanistic driver is earlier on given its importance to the main message of the paper.

Table 1a: Suggest adding LA-ICP-MS in the caption. It costs nothing and it's helpful to know immediately that these are laser data. One of the headings says "Number of data". Are these also numbers of grains or numbers of analyses/dates and some grains were hit multiple times?

Lines 166-167: I was pleased to see the authors also accounted for sufficient time for minimum subduction initiation to melt generation. Seems this also requires some knowledge about subduction angle?

Figure 6: Check that acronyms/abbreviations are defined in the caption (e.g., A-ARC and TA).

Lines 377 to 380: I'm not sure I follow this logic. The authors note that previous studies using mercury data (a volcanism

proxy) did not support NAIP volcanism as sole mechanism for PETM... if true, how does this support the case for the role of Aleutian volcanism? Why wouldn't Aleutian volcanism contribute to a mercury signal?

Reviewer #2

(Remarks to the Author)

Manuscript ID: NCOMMS-25-30203

Title: Aleutian Arc Initiation at 56 Ma: Tectonic and climatic implications

Recommendation: Major Revision

Overall Assessment

This study presents valuable Ar/Ar and U/Pb geochronology and whole-rock geochemical data (major/trace elements, Sr-Nd-Pb isotopes) from lavas and granodiorites across the western Aleutian arc. The work leverages rare samples from international ocean-drilling expeditions to argue that Aleutian subduction initiated ≥ 56 Ma, triggering a 10-Ma circum-Pacific plate reorganization with potential links to the EECO (Early Eocene Climatic Optimum) and ETM (Paleocene-Eocene Thermal Maximum). The findings are novel and align with NC's scope, but significant revisions—particularly in data presentation and contextualization—are required before acceptance.

Major Comments

1. Clarify Climate Linkages

Abstract: It is recommended that in the abstract section, the relationship between Siletzia LIP volcanism and the EECO be emphasized.

Discussion: Expand on how subduction-driven plate reorganization influenced Siletzia volcanism and its potential climate impacts (PETM/EECO). Add 1–2 sentences linking mantle dynamics to surface environmental change.

2. Add Critical Comparative Figure

In "Aleutian Subduction Initiation" (Discussion), include a schematic cross-section comparing the Aleutian and Izu-Bonin-Mariana (IBM) arcs during their initial subduction phases. Annotate with: Key lithologies, Age ranges of volcanic sequences, Tectonic settings, etc.

Rationale: This visual is vital for readers to assess the uniqueness of Aleutian subduction initiation versus established Pacific models.

Essential Revisions

Figures & Captions

1. Fig. 1, Missing panel labels, Add "A" and "B" to subplots.
2. All figures, Inconsistent font sizes, Standardize labels/legends to 10–12 pt.
3. Fig. 6, Abbreviation mismatch (A-ARC vs. "AI" in caption).
4. Fig. 6, Hotspots not plotted, Add hotspots (e.g., **★**) as referenced on p.10, line 27.

Text & Terminology

1. Figure citations: Correct erroneous "Fig. 6" references (e.g., p.9 line 10; p.9 line 45; p.10 line 10; p.10 line 27).

Recommendations for Strengthening

1. Highlight sample scarcity: Add to Abstract/Introduction:

"Samples acquired via international ocean drilling provide unique constraints on the Aleutian arc's early evolution."

2. Temper causal claims: Replace definitive phrasing (e.g., "caused PETM") with:

"may have contributed to PETM warming."

3. New figure design: For the IBM-Aleutian comparison figure:

- Use simplified stratigraphic columns annotated with ages.

- Cite data sources for IBM lithologies (e.g., Reagan et al., 2019 GSA Bulletin).

Reviewer #3

(Remarks to the Author)

Review of "Aleutian Arc Initiation at 56 Ma: Tectonic and climatic implications" by Hoernle et al., submitted for consideration of publication in Nature Communications

This manuscript presents new geochronological and geochemical data from four locations along the Aleutian magmatic arc, showing that magmatism related to the Aleutian subduction zone was already active by ~ 56 Ma. These ages are nearly 10 Ma older than the oldest reliable $^{40}\text{Ar}/^{39}\text{Ar}$ ages previously reported from the Aleutian arc, extending the history of the Aleutian arc and subduction zone back into the Paleocene. These results have important implications for the tectonic history of the northern Pacific, and potentially for the broader Pacific realm. The authors integrate these new data into a revised plate tectonic reconstruction for the Pacific region, linking the onset of subduction to a change in absolute plate motion of the Pacific plate around 56 Ma. This plate motion change is interpreted within the context of a ~ 10 -Ma-long plate reorganization in the Pacific realm, culminating in the formation of the enigmatic Hawaii-Emperor Bend. In addition, this study proposes a connection between these tectonic changes and major climatic events, including the Paleocene-Eocene Thermal Maximum (PETM) and the Early Eocene Climate Optimum (EECO). The new age and geochemical data presented in this study provide important new constraints on the tectonic history of the Aleutian subduction zone, and therefore deserve to be

published. However, the proposed connections between northern Pacific tectonics and climatic and biotic events remain speculative. In its current form, the manuscript does not provide clear evidence for such causal relationships. In my opinion, the discussion of the climatic impacts should be limited to what can really be inferred from the results presented here and framed with appropriate caution. In addition, this article would benefit from a sharper focus on the tectonic implications of the important new data and on updating the tectonic model for the early Cenozoic history of the northern Pacific. I therefore recommend major revision of the current manuscript prior to publication. Below, I provide major and minor comments that the authors can use to revise and strengthen their manuscript.

Major comments

First, essential background on the current understanding of the tectonic history of the Aleutian subduction zone and arc is missing. What are recent ideas on the age and cause of subduction zone initiation? Without this context, it is difficult to adequately interpret the geochronological and geochemical results for any reader that has no detailed knowledge on the Aleutian subduction zone. While the oldest age determinations are quoted, no summary is presented of recent models of how and when the Aleutian subduction zone started. Important papers from the 1980s are cited in the introduction (refs 9 to 12), but more recent hypotheses related the origin of the Aleutian arc, presented e.g., in Scholl (2007, *Geophys. Monogr. Ser.*), Vaes et al. (2019, *Tectonics*), Cramer et al. (2020, *Nat. Comm.*), Stern et al. (2025, *Int. Geol. Rev.*), are not discussed. For instance, the authors simply mention their preference for the plate capture model for the Aleutian Basin origin in lines 274-275, without providing any explanation on the two competing models presented in Stern et al. (2025). This likely makes the paper difficult to follow readers without much background on northern Pacific tectonics.

The authors revised a widely used global plate model (of Müller et al., 2019) that integrates the new constraints. I would expect a more detailed description of the construction of this model and how this relates to and provides an improvement over previous plate-kinematic reconstructions, e.g., from Domeier et al. (2017, *Sci. Adv.*), Vaes et al. (2019, *Tectonics*), Hu et al. (2022, *Nat. Geosc.*) and Calvelage et al. (2024, *Proc. R. Soc. A*). Importantly, the GPlates reconstruction files were not made available, severely limiting the future use of this revised plate model. The context of important aspects of the reconstruction snapshots shown in Fig. 6 are often unclear. Which marine magnetic anomaly data (?) provides evidence for the changes in plate velocity (at 56 Ma) and direction of the Pacific plate (at 55 and 52 Ma)? Are these significant, i.e., not within uncertainty of the Euler rotation poles? The absolute reference frame used to quantify the direction and velocity of all plate motions shown in Fig. 6 is also not mentioned. This makes it impossible to assess whether these changes in Pacific plate motion are related to (uncertainties in) the mantle reference frame or a true signal of a change in relative plate motion linked to changes in torques generated by slab pull. In any case, it may be difficult to imagine that the relatively short western segment of the Aleutian subduction zone could pull the entire Pacific plate in a different direction at ~55 Ma. Domeier et al. (2017, *Sci. Adv.*) argued the exact opposite: that the newly formed north-dipping Aleutian subduction zone would generate less slab pull, increasing the relative contribution of the westward-dipping subduction zones along the western Pacific margins. This must be clarified.

To make it much easier to understand the proposed tectonic changes in the north Pacific region (lines 260-275), including the hypothesized initiation of subduction along a pre-existing transform fault and/or fracture zone, I suggest adding a figure that either shows a zoomed-in version of the tectonic evolution of this region or a conceptual model of it. The formation of the Aleutian subduction zone at ~56 Ma is not straightforward to interpret from figures 6b and c. Also, the proposed links with the Bowers and Shirshov Ridges, as well as the implications for potential plate capture below the Aleutian Basin, are difficult to follow without a figure that visualizes the tectonic evolution described in this paper. This would also help to understand which oceanic lithosphere would be captured in the proposed tectonic model, and why the hypothesized formation of the Aleutian subduction zone is consistent with N-S magnetic lineations in the Aleutian Basin.

Finally, I strongly recommend the authors to revise their interpretations and discussion related to the potential climatic impact of the tectonic changes in the northern Pacific region. Although these new age data indeed show that the initiation of subduction in the Aleutian arc may be (nearly) synchronous to the PETM, it is very speculative to suggest any causal relationship. The ~56-55 Ma age data provide a minimum age of subduction-related magmatism. It may very well be that future dating of new samples will push this age some million years further back in time. More importantly, it is not clear to me why the earliest stage of Aleutian arc magmatism would have triggered a massive release of greenhouse gases sufficient to drive the exceptionally fast (and short-lived!) warming associated with the PETM. If the initiation of a subduction zone and arc was an important climate driver, one may expect similar hyperthermal events to accompany the onset of other subduction zones like the IBM and Tonga-Kermadec. It is unclear why a shift in subduction from the Kamchatka/Koryak margin to the Aleutians would produce such a pronounced climatic impact, including a major carbon isotope excursion. In my view, the current evidence for a causal relationship is not strong enough to justify the inclusion of a figure like Fig. 8. Regarding longer-term climatic impacts, I find Fig. 7 too heavily focused on the tectonic and magmatic events in the Pacific realm. Apart from the ~62 Ma onset of NAIP magmatism and the initial India-Asia initial, key events outside of the Pacific, such as the large early Eocene flare-up of the Gangdese arc, are notably absent. To adequately address long-term climate change in the early Cenozoic, a more comprehensive and global framework is needed. Attempting to explain major climatic and biotic events primarily through Pacific tectonic changes is likely too narrow and speculative.

In summary, I recommend substantially shortening the discussion on the climatic impacts of the Pacific-wide plate reorganization and placing more emphasis on the tectonic implications of the new data and the reconstruction of northern Pacific tectonics. Outlining potential connection between Pacific tectonics and early Cenozoic climate change could still provide a useful platform for future research, as long as these are supported by the actual data and framed with appropriate caution.

Minor comments:

Line 21: A plate tectonic reconstruction does not constrain the initiation of subduction. It is the data that provide those constraints. The reconstruction is a model that builds upon and integrates these data. I would say that the main contribution here is the data. There may be various ways to explain these data in terms of plate tectonic reconstructions.

Line 40: Replace “haven’t” by “have not”

Line 64-68: This sentence is overly long and could be split into two and expanded. I suggest providing more details in terms of the nature and timing of key events, supported by relevant references that describe those specific events and to papers on the Pacific-wide plate reorganization, like Whittaker et al. (2007, Science), Seton et al. (2015, GRL), Vaes et al. (2019, Tectonics), Hu et al. (2022, Nat. Geosc.) etc. (some of which are cited later on in the manuscript).

Lines 69-74: The links between the regional tectonics of the northern Pacific and these climate and environmental ‘events’ are not evident here and no references are provided to previous articles providing such links. I suggest removing these sentences here and leaving the discussion of potential links between Pacific tectonics and climate to the Discussion section.

Lines 74-75: There is no widely accepted explanation for the PETM, but saying that there is no ‘viable’ explanation would be overly dismissive of the large body of work that has been done on the PETM, e.g., on large methane release, and magmatic intrusions into organic-rich sediments linked to the North Atlantic Large Igneous Provinces. Again, no references are provided to support this claim. Surprisingly, a discussion of potential triggers is included in the final section of the manuscript, which does not seem consistent with this strong statement.

Line 76: It reads now as if the question of what caused the PETM can be directly addressed by new age and geochemical data from the western Aleutian arc. I reiterate that it may be better to only mention in the Introduction that the climatic implications of the new results will be discussed later on in the article.

Lines 78-83: Reads like a very short geological background. As mentioned above, this should be expanded to a summary that describes key geological and tectonic constraints on the evolution of the Aleutian subduction zone and arc.

Lines 149-156: This is an important result, arguing against several previous models that linked the Komandorsky Block to the Kronotsky arc. Are the geochemical signatures also different from the early Cenozoic Kronotsky arc rocks? I suggest showing in Fig. 3, or at least mentioning, the difference in geochemical signature.

Line 152: the paleomagnetic data of Upper Cretaceous rocks attributed to the Kronotsky arc yield a paleolatitude of ~30° (Harbert et al., 2009 SMSPS), which I would not refer to as ‘equatorial’.

Lines 161-163: Davis et al. (1989), cited here and in Fig. 1, obtained 54.4-50.2 Ma K-Ar ages from magmatic rocks of the Beringian margin, leading e.g. Scholl (2007) and Vaes et al. (2019) to assume a ~50 Ma jump of subduction to the incipient Aleutian subduction zone. Any ideas why magmatism would still be active there?

Lines 164-165: It is not clear why a very high convergence velocity of 10-20 cm/yr is assumed, based on the two old papers cited here. Why not derive this velocity from the GPlates plate model?

Lines 211-212: The spontaneous nature of the initiation of the IBM subduction zone is still debated (see e.g., Cramer et al. 2020 Nat. Comm., Liu et al. 2024 Comm. Earth & Env., van de Lagemaat et al. 2024 GR). Regardless of whether the authors support the spontaneous subduction initiation model, it would be fair to mention that it is still a topic of active research.

Line 217: The ages quoted here are not entirely correct if derived from Vaes et al. (2019). This paper, referenced here, estimated the initiation of the Olyutorsky arc at ~85 Ma and showed that the collision with Kamchatka, although indeed starting at 60-55 Ma, was diachronous, with obduction of its southern segment at ~55-50 Ma and of its northern segment at ~50-45 Ma. This is described further on in the manuscript (lines 261-264), so why not already mention that here?

Lines 245-247 and Fig 6c: Why would the Pacific plate motion change to almost due north when the Izanagi-Pacific ridge just subducted and subduction of the actual Pacific plate below East Asia also initiated at around 56 Ma? It is not clear what is meant with the statement that the change (clockwise rotation?) in plate motion is also observed when keeping West Antarctica fixed. I assume the authors mean that the change is visible in the marine magnetic anomalies, but without showing that this remains difficult to follow.

Lines 300-302 and 315-316: It is not straightforward to see how the slab pull of the Kronotsky subduction zone could have affected Pacific plate motion in a scenario where the Kula plate, and not the Pacific plate, subducts below the Kronotsky arc (Fig. 6). The youngest history of the Kronotsky arc discussed here is confusing: the authors mention the ~42 Ma estimate for the cessation of the arc, but at the same time refer to a 47 Ma slab break-off and cessation of northward-dipping subduction (Fig. 6f). I do not understand how there is a polarity reversal when subduction stops. Figure 6 also does not show any southward-dipping subduction post-55 Ma anywhere in the northern Pacific.

Lines 320-321: Which hotspot reference frame?

Lines 327-330: It is clear that many major tectonic changes occurred in this time interval. Given that the authors focus a lot on the timing of these events, wouldn’t it be helpful to place these sequential events in the framework of a tectonic chain reaction (Gürer et al., 2022 Nat. Geosc.)?

Data availability: The link to the Dryad repository did not work on my computer; it did not seem active. I advise to double-check this, as open access to the data is essential for future scrutiny of the data.

Bram Vaes

Version 1:

Reviewer comments:

Reviewer #1

(Remarks to the Author)

Review of Hoernle et al: Aleutian arc initiation at 56 Ma: Tectonic and climatic implications

The authors have sufficiently addressed my previous concerns. I provide a few additional comments/concerns that if addressed may strengthen the manuscript, wherever it is published. This is an important dataset which constrains the timing of initiation of the Western Aleutian Arc and informs regional Pacific plate tectonic models.

Lines 32-48 should be supported by references. Especially the discussion of previous geochronological constraints. Perhaps these are refs 1-8 as the next paragraph begins by citing reference 9?

The authors motivate the importance/impact of the new geochronology by emphasizing that previous ~55Ma ages are "unreliable" because they've not been reproduced since and because the uncertainties are large (lines 39-42). I do not think this necessarily makes them 'unreliable' does it? A date can be imprecise but still accurate. In the spirit of critical review, could one argue that these new results more precisely confirm, and are consistent with, previous constraints?

Line 74: HEB needs to be defined.

Line 78: Why the TF and FZ acronyms?

Lines 552-554 need citation(s)

I like the addition of Figure 4. Expanding panels e-h to show Beringian Margin subduction until ~50 Ma (Davis et al., 1989) would be more comprehensive and better support (somewhat) the authors' description/interpretation of their preferred model for western Aleutian initiation (as well as text in lines 327-329).

The manuscript needs to be checked throughout for use, placement, and defining of acronyms as there are instances where they are not defined, redefined, and/or perhaps not necessary.

Reviewer #2

(Remarks to the Author)

Thank you for the thorough and thoughtful revisions made to your manuscript in response to the previous round of review. I have carefully examined the revised version and your detailed point-by-point responses. I am pleased to note that you have addressed all of my major concerns comprehensively and effectively.

The modifications to the text, the additional data included (particularly regarding comparison of Izu-Bonin-Mariana (IBM) with Aleutian and Kamchatka subduction/arc initiation), and the refined interpretations have significantly strengthened the manuscript. The arguments are now more robust, the conclusions are well-supported by the evidence presented, and the overall clarity has been improved.

In its current form, the work presents a noteworthy contribution to the field through rare samples from lavas and granodiorites from four submarine basement sequences spanning the western Aleutian Arc. The methodology is sound, the results are compelling, and the discussion appropriately contextualizes the findings within the established literature.

I believe the manuscript now meets the high standards expected for publication in Nature Communications. I have no further substantive criticisms and recommend its acceptance without further revision.

I congratulate the authors on a fine piece of work.

Best regards,

Zhao Lei

Reviewer #3

(Remarks to the Author)

First, I would like to compliment the authors for their thorough revisions and detailed responses to all points raised by the reviewers. In my opinion, the decision to focus the manuscript on the tectonic implications of their new data, rather than speculative climatic impacts, has significantly strengthened its scientific value. The revised manuscript represents a clear and important advance in our understanding of the complex tectonic history of the northern Pacific region. By integrating recent tectonic models with their new geochemical and age constraints, the authors provide a robust and comprehensive model for the initiation and early evolution of the Aleutian subduction zone and arc. Moreover, the addition of Figures 4 and 5 provide strong visual support for the tectonic model and interpretations of the data. The authors now also discuss enigmatic features of this region, like the Shirshov and Bowers Ridges, in more detail and highlight some outstanding questions and problems for future research. I consider this new tectonic model for the Aleutian arc and broader Bering Sea region as the state-of-the-art and am confident it will serve as a platform for further research. Finally, the authors provide a more nuanced discussion of the potential links between the Aleutian arc initiation and climatic events like the PETM, carefully outlining possible mechanisms. I therefore recommend this manuscript for publication in Nature Communications, after addressing the (very) minor revisions suggested below.

Title: It may be good to indicate the uncertainty in the age of arc initiation, e.g., by writing '~56 Ma'.

Introduction: the tectonic models discussed here built on important work of Russian geologists. I realize that the number of references is limited but citing some key works on arc-continent collision in Kamchatka like Konstantinovskaya (2001, Tectonophys.) and Shapiro & Soloviev (2009, Russ. Geol. Geophys.) would help acknowledging this.

Lines 32-48: The references to published ages (1-8) seem to be missing in the revised manuscript.

Line 175: 'was not accreted until the mid-Miocene'

Caption Figure 4: Add that Manus plume 'may have caused uplift'?

Line 476: replace 'no doubt' by 'undoubtedly'

Lines 548-551: This is a very long and complicated sentence, I would suggest splitting it up and rewriting it.

Bram Vaes

REVIEWER COMMENTS

Reviewer #1 (Remarks to the Author):

Hoernle and others present new geochronology and geochemistry from dredged arc rocks in the western Aleutians to provide new constraints on the initiation of Aleutian arc volcanism and subduction. Their findings (initiation >56 Ma) are consistent with previously proposed models linking Aleutian subduction with cessation of Alaska Range and Siberian/Koryak volcanism. Overall, the paper is well-written and its intended message is clear/consistent. The study provides novel and important data/results from a remote geologic setting. Moreover, as highlighted by the authors, the geodynamic initiation and history of Aleutian arc is a critical piece to regional Pacific plate reconstructions, which is well-supported.

New results convincingly demonstrate that western Aleutian volcanism began by 56 Ma, resolving any previous uncertainties between older initiation (56-50 Ma) and younger initiation based on more reliable direct arc dating (<50 Ma). The authors interpret (or propose) a causal relationship between coeval Pacific plate reorganizations with warming of the PETM. As I understand it, their argument is that enhanced volcanism associated with these events tipped the scales of climate response against a backdrop of already enhanced volcanism associated with the North Atlantic Igneous Province (NAIP). Although I have no objections to the possibility of these mechanisms, my main concern is that it is based primarily on correlation. Because plate reorg in and of itself is not an actual mechanism, the authors propose that this reorg results in increased global volcanism. This is perhaps the main area of concern in the paper, in its current form. To make a more convincing argument, it would help to show that this plate reorg results in a marked increase in volcanism (or emitted CO₂) at that time. For example, could they total the length of Pacific subduction zone interfaces before and after this plate reorg to show an abrupt change (increase) in volcanism. If not that, then something tangible to better connect a link between how this plate reorg results in substantial increases in CO₂. Moreover, it would also help to clarify if (and why) there's a net gain in volcanism between the loss of the Olyutorsky Arc and the initiation of the western Aleutian Arc (lines 381-382: "it provides an additional large-scale potential carbon source in the North Pacific for this global event").

A causal relationship is conceivable through the sudden release of large amounts of carbon due to intrusion of arc magmas into organic-rich sediments on the seafloor or a tectonic event, such as uplift of a large epicontinental seaway. We have added these possible mechanisms to the discussion, despite shortening the possible link to the PETM significantly, since this is not the main direction of the manuscript.

Lines 39 to 41: Something is off (typo?) in these two sentences. Important details are placed into a string of hard-to-follow parentheses. I suggest just placing this into a simpler sentence format.

Have done.

Line 74-75: Authors state: "To date, there is no viable explanation for the cause of the PETM." Perhaps there is no uniform consensus on the cause(s) for the PETM but multiple 'viable' mechanisms have been proposed, including those discussed later in this paper.

Have modified the text accordingly.

Line 76: Paragraph begins with "To address these questions", however no questions were presented in the previous paragraph. I suggest either posing some questions or clarifying what is being addressed (and how) by presenting new geochronology and whole rock data (e.g., uncertainties, controversies, gaps in knowledge, etc.).

Have revised the text.

The link between Aleutian Arc initiation and the PETM is alluded to multiple times early in the manuscript (abstract, lines 70-72, lines 90-91) before the reader gets to the actual proposed causal mechanism (increased C-rich volcanism). Consider a least a brief mention of what the proposed mechanistic driver is earlier on given its importance to the main message of the paper.

In accordance with the editor's and reviewer 3's recommendation to focus primarily on Aleutian subduction initiation and the Eocene plate reorganization, we have reduced the text pertaining to the climatic events significantly and therefore do not add additional details earlier in the manuscript.

Table 1a: Suggest adding LA-ICP-MS in the caption. It costs nothing and it's helpful to know immediately that these are laser data. One of the headings says "Number of data". Are these also numbers of grains or numbers of analyses/dates and some grains were hit multiple times?

Have added LA-ICP-MS and clarified number of analyses.

Lines 166-167: I was pleased to see the authors also accounted for sufficient time for minimum subduction initiation to melt generation. Seems this also requires some knowledge about subduction angle?

These are estimates from the literature that are consistent with the kinematics of the G-plates model that we present in the manuscript. We cannot constrain the subduction angle; the time we give is the minimum age.

Figure 6: Check that acronyms/abbreviations are defined in the caption (e.g., A-ARC and TA).

They are now defined in this figure caption.

Lines 377 to 380: I'm not sure I follow this logic. The authors note that previous studies using mercury data (a volcanism proxy) did not support NAIP volcanism as sole mechanism for PETM... if true, how does this support the case for the role of Aleutian volcanism? Why wouldn't Aleutian volcanism contribute to a mercury signal?

Mercury serves as a proxy for volcanic activity. The sediment cores were taken in the North Atlantic close to areas of NAIP volcanism, but none of the cores showed a mercury spike indicative of volcanism at the beginning of the PETM. North Atlantic cores would not record a signal from volcanism in the Pacific. We have clarified in the manuscript.

Reviewer #2 (Remarks to the Author):

Manuscript ID: NCOMMS-25-30203

Title: Aleutian Arc Initiation at 56 Ma: Tectonic and climatic implications

Recommendation: Major Revision

Overall Assessment

This study presents valuable Ar/Ar and U/Pb geochronology and whole-rock geochemical data (major/trace elements, Sr-Nd-Pb isotopes) from lavas and granodiorites across the western Aleutian arc. The work leverages rare samples from international ocean-drilling expeditions to argue that Aleutian subduction initiated ≥ 56 Ma, triggering a 10-Ma circum-Pacific plate reorganization with potential links to the EECO (Early Eocene Climatic Optimum) and ETM (Paleocene-Eocene Thermal Maximum). The findings are novel and align with NC's scope, but significant revisions—

particularly in data presentation and contextualization—are required before acceptance.

Major Comments

1. Clarify Climate Linkages

Abstract: It is recommended that in the abstract section, the relationship between Siletzia LIP volcanism and the EECO be emphasized.

Based on reviewer 3 and editor's suggestion, we have reduced the discussion of the climate linkages. We no longer discuss the EECO in the manuscript now and primarily refer to possible linkages with the PETM and the Cenozoic hothouse as a topic that should be looked at in more detail.

Discussion: Expand on how subduction-driven plate reorganization influenced Siletzia volcanism and its potential climate impacts (PETM/EECO). Add 1–2 sentences linking mantle dynamics to surface environmental change.

As noted above, we no longer discuss EECO or the link of Siletzia volcanism to the EECO and have shortened our discussion of possible climatic implications.

2. Add Critical Comparative Figure

In "Aleutian Subduction Initiation" (Discussion), include a schematic cross-section comparing the Aleutian and Izu-Bonin-Mariana (IBM) arcs during their initial subduction phases. Annotate with: Key lithologies, Age ranges of volcanic sequences, Tectonic settings, etc.

Rationale: This visual is vital for readers to assess the uniqueness of Aleutian subduction initiation versus established Pacific models.

We find this a very good suggestion by the reviewer and have added the recommended figure (figure 4 in the manuscript now).

Essential Revisions

Figures & Captions

1. Fig. 1, Missing panel labels, Add "A" and "B" to subplots.

Have added these.

2. All figures, Inconsistent font sizes, Standardize labels/legends to 10–12 pt.

We have standardized labels/labels wherever possible.

3. Fig. 6, Abbreviation mismatch (A-ARC vs. "A" in caption).

We now use Aleu in both figure and caption.

4. Fig. 6, Hotspots not plotted, Add hotspots (e.g., **★**) as referenced on p.10, line 27.

Added Hawaiian hotspot location.

Text & Terminology

1. Figure citations: Correct erroneous "Fig. 6" references (e.g., p.9 line 10; p.9 line 45; p.10 line 10; p.10 line 27).

The numbering of figures has changed, since some have been removed and we have added the new figure that this reviewer recommended. We have checked all figure references.

Recommendations for Strengthening

1. Highlight sample scarcity: Add to Abstract/Introduction:

"Samples acquired via international ocean drilling provide unique constraints on the Aleutian arc's early evolution."

We have added this information to the Abstract and Introduction, but would like to note that these samples were not obtained by IODP drilling but rather through dredging at the base of the Aleutian Arc and through sampling of basal complex on Medny Island. We appreciate that the reviewer recognizes that these are very rare samples.

2. Temper causal claims: Replace definitive phrasing (e.g., "caused PETM") with: "may have contributed to PETM warming."

Have done so.

3. New figure design: For the IBM-Aleutian comparison figure:

- Use simplified stratigraphic columns annotated with ages.
- Cite data sources for IBM lithologies (e.g., Reagan et al., 2019 GSA Bulletin).

We do not compare stratigraphic columns because the variation in compositions at the Aleutians is spatial rather than temporal as is the case for the IBM. Therefore, we show the evolution of both arcs in the form of cartoons at different ages, including the information requested by this reviewer previously. We cite Reagan et al (2019 and 2023).

Reviewer #3 (Remarks to the Author):

Review of "Aleutian Arc Initiation at 56 Ma: Tectonic and climatic implications" by Hoernle et al., submitted for consideration of publication in Nature Communications

This manuscript presents new geochronological and geochemical data from four locations along the Aleutian magmatic arc, showing that magmatism related to the Aleutian subduction zone was already active by ~56 Ma. These ages are nearly 10 Ma older than the oldest reliable $^{40}\text{Ar}/^{39}\text{Ar}$ ages previously reported from the Aleutian arc, extending the history of the Aleutian arc and subduction zone back into the Paleocene. These results have important implications for the tectonic history of the northern Pacific, and potentially for the broader Pacific realm. The authors integrate these new data into a revised plate tectonic reconstruction for the Pacific region, linking the onset of subduction to a change in absolute plate motion of the Pacific plate around 56 Ma. This plate motion change is interpreted within the context of a ~10-Ma-long plate reorganization in the Pacific realm, culminating in the formation of the enigmatic Hawaii-Emperor Bend. In addition, this study proposes a connection between these tectonic changes and major climatic events, including the Paleocene-Eocene Thermal Maximum (PETM) and the Early Eocene Climate Optimum (EECO). The new age and geochemical data presented in this study provide important new constraints on the tectonic history of the Aleutian subduction zone, and therefore deserve to be published. However, the proposed connections between northern Pacific tectonics and climatic and biotic events remain speculative. In its current form, the manuscript does not provide clear evidence for such causal relationships. In my opinion, the discussion of the climatic impacts should be limited to what can really be inferred from the results presented here and framed with appropriate caution. In addition, this article would benefit from a sharper focus on the tectonic implications of the important new data and on updating the tectonic model for the early Cenozoic history of the northern Pacific. I therefore recommend major revision of the current manuscript prior to publication. Below, I provide major and minor comments that the authors can use to revise and strengthen their manuscript.

Major comments

First, essential background on the current understanding of the tectonic history of the Aleutian subduction zone and arc is missing. What are recent ideas on the age and cause of subduction zone initiation? Without this context, it is difficult to adequately interpret the geochronological and geochemical results for any reader that has no detailed knowledge on the Aleutian subduction zone. While the oldest age determinations are quoted, no summary is presented of recent models of how and when the Aleutian subduction zone started. Important papers from the 1980s are cited in the introduction (refs 9 to 12), but more recent hypotheses related the origin of the Aleutian arc, presented e.g., in Scholl (2007, Geophys. Monogr. Ser.), Vaes et al. (2019, Tectonics), Crameri et al. (2020, Nat. Comm.), Stern et al. (2025, Int. Geol. Rev.), are not discussed. For instance, the authors simply mention their preference for the plate capture model for the Aleutian Basin origin in lines 274-275, without providing any explanation on the two competing models presented in Stern et al. (2025). This likely makes the paper difficult to follow readers without much background on northern Pacific tectonics.

We have addressed these comments in our revision, summarizing and citing the recent studies recommended by the reviewer. Plate capture model vs backarc spreading model is now discussed in more detail in the manuscript, as well as our reasons for favoring the plate capture model.

The authors revised a widely used global plate model (of Müller et al., 2019) that integrates the new constraints. I would expect a more detailed description of the construction of this model and how this relates to and provides an improvement over previous plate-kinematic reconstructions, e.g., from Domeier et al. (2017, Sci. Adv.), Vaes et al. (2019, Tectonics), Hu et al. (2022, Nat. Geosc.) and Calvelage et al. (2024, Proc. R. Soc. A). Importantly, the GPlates reconstruction files were not made available, severely limiting the future use of this revised plate model. The context of important aspects of the reconstruction snapshots shown in Fig. 6 are often unclear. Which marine magnetic anomaly data (?) provides evidence for the changes in plate velocity (at 56 Ma) and direction of the Pacific plate (at 55 and 52 Ma)? Are these significant, i.e., not within uncertainty of the Euler rotation poles? The absolute reference frame used to quantify the direction and velocity of all plate motions shown in Fig. 6 is also not mentioned. This makes it impossible to assess whether these changes in Pacific plate motion are related to (uncertainties in) the

mantle reference frame or a true signal of a change in relative plate motion linked to changes in torques generated by slab pull. In any case, it may be difficult to imagine that the relatively short western segment of the Aleutian subduction zone could pull the entire Pacific plate in a different direction at ~55 Ma. Domeier et al. (2017, *Sci. Adv.*) argued the exact opposite: that the newly formed north-dipping Aleutian subduction zone would generate less slab pull, increasing the relative contribution of the westward-dipping subduction zones along the western Pacific margins.

We have added a more detailed description of the revised plate model at the end of the Methods section that addresses these questions. We also now make the revised plate model available.

Domeier et al. (2017) believe that Aleutian SZI took place at ~46 Ma, based on the oldest published age from the Aleutians from Jicha et al. (2006). They write: "Equally puzzling is the apparent onset of north-dipping subduction of the Pacific plate beneath the Aleutian arc (Fig. 1) at HEB time (Jicha et al., 2006), which, intuitively, should have presented a northward pull at approximately the time that the plate motion switched from north-directed to northwest-directed." We completely agree and note that our new dates place Aleutian SZI 10 Ma earlier and thus there is no direct/conflicting link to the HEB. In our model, accretion of the Olyutorsky Arc causes occlusion of that subduction system and subduction of Izanagi-Pacific Ridge causes a temporary cessation of subduction in the western Pacific until the IBM Arc system is established. At the same time, formation and accretion of the Siletzia and Yakutat LIPs to western North America between 56-53 Ma slowed/shut down subduction in the east. Aleutian subduction, coupled with subduction beneath the Beringian margin (until ~50 Ma) and subduction beneath Alaska would have allowed subduction to the north.

To make it much easier to understand the proposed tectonic changes in the north Pacific region (lines 260-275), including the hypothesized initiation of subduction along a pre-existing transform fault and/or fracture zone, I suggest adding a figure that either shows a zoomed-in version of the tectonic evolution of this region or a conceptual model of it. The formation of the Aleutian subduction zone at ~56 Ma is not straightforward to interpret from figures 6b and c. Also, the proposed links with the Bowers and Shirshov Ridges, as well as the implications for potential plate capture below the Aleutian Basin, are difficult to follow without a figure that visualizes the tectonic evolution described in this paper. This would also help to understand which oceanic lithosphere would be captured in the proposed tectonic model, and why the hypothesized formation of the Aleutian subduction zone is consistent with N-S magnetic lineations in the Aleutian Basin.

We have added a conceptual blow-up model of the tectonic evolution of the northern Pacific region to make it easier to follow the text. The conceptual model however does not cover the Bowers Ridge, which formed in the Eocene, and is not a main part of Aleutian Arc initiation or the 10 Ma plate tectonic reorganization. Nevertheless, we have tried to clarify the text pertaining to Bowers further. Finally, we note that it is not clear if the N-S magnetic lineations in the Aleutian Basin actually represent seafloor spreading anomalies, and although we have incorporated it into the model (following Vaes et al. (2019), we do not think that too much weight should be placed on explaining them as seafloor spreading anomalies.

Finally, I strongly recommend the authors to revise their interpretations and discussion related to the potential climatic impact of the tectonic changes in the northern Pacific region. Although these new age data indeed show that the initiation of subduction in the Aleutian arc may be (nearly) synchronous to the PETM, it is very speculative to suggest any causal relationship. The ~56-55 Ma age data provide a minimum age of subduction-related magmatism. It may very well be that future dating of new samples will push this age some million years further back in time. More importantly, it is not clear to me why the earliest stage of Aleutian arc magmatism would have triggered a massive release of greenhouse gases sufficient to drive the exceptionally fast (and short-lived!) warming associated with the PETM. If the initiation of a subduction zone and arc was an important climate drive, one may expect similar hyperthermal events to accompany the onset of other subduction zones like the IBM and Tonga-Kermadec. It is unclear why a shift in subduction from the Kamchatka/Koryak margin to the Aleutians would produce such a pronounced climatic impact, including a major carbon isotope excursion. In my view, the current evidence for a causal relationship is not strong enough to justify the inclusion of a figure like Fig. 8. Regarding longer-term climatic impacts, I find Fig. 7 too heavily focused on the tectonic and magmatic events in the Pacific realm. Apart from the ~62 Ma onset of NAIP magmatism and the initial India-Asia initial, key events outside of the Pacific, such as the large early Eocene flare-up of the Gangdese arc, are notably absent. To adequately address long-term climate change in the early Cenozoic, a more comprehensive and global framework is needed. Attempting to explain major climatic and biotic events primarily through Pacific tectonic changes is likely too narrow and speculative.

In summary, I recommend substantially shortening the discussion on the climatic impacts of the Pacific-wide plate reorganization and placing more emphasis on the tectonic implications of the new data and the reconstruction of northern Pacific tectonics. Outlining potential connection between Pacific tectonics and early Cenozoic climate change could still provide a useful platform for future research, as long as these are supported by the actual data and frame with appropriate caution.

We agree with the reviewer and have substantially shortened the discussion on the climatic impacts of the Pacific-wide plate reorganization placing more emphasis on the tectonic implications. We have also removed figures 7 and 8 in this version. We do however think that Aleutian SZI and IBM, Tonga, etc. SZI in the Pacific realm during the plate tectonic reorganization should be considered in explaining the PETM and the Cenozoic hothouse.

Minor comments:

Line 21: A plate tectonic reconstruction does not constrain the initiation of subduction. It is the data that provide those constraints. The reconstruction is a model that builds upon and integrates these data. I would say that the main contribution here is the data. There may be various ways to explain these data in terms of plate tectonic reconstructions.

We have changed the text accordingly.

Line 40: Replace “haven’t” by “have not”

Have done.

Line 64-68: This sentence is overly long and could be split into two and expanded. I suggest providing more details in terms of the nature and timing of key events, supported by relevant references that describe those specific events and to papers on the Pacific-wide plate reorganization, like Whittaker et al. (2007, Science), Seton et al. (2015, GRL), Vaes et al. (2019, Tectonics), Hu et al. (2022, Nat. Geosc.) etc. (some of which are cited later on in the manuscript).

Have rewritten. Also, as recommended previously by this reviewer, we now include a detailed review of more recent models for the Aleutian subduction initiation.

Lines 69-74: The links between the regional tectonics of the northern Pacific and these climate and environmental ‘events’ are not evident here and no references are provided to previous articles providing such links. I suggest removing these sentences here and leaving the discussion of potential links between Pacific tectonics and climate to the Discussion section.

We have reduced the text here and added reference to Reagan et al. (2013), who provide such a link (e.g. last three sentences of their abstract): “The volume of basalt erupted near western Pacific trenches associated with subduction initiation and early-arc development in the early Eocene could rival the volumes of large igneous provinces. The eruption of these basalts corresponds with the height of the Early Eocene Climatic Optimum (EECO), when global atmospheric temperatures were likely at or near their Cenozoic maximum. Therefore, CO₂ vented during this volcanism, as well as that associated with the North Atlantic Igneous Province, the Siletzia terrane, and slab rollback and detachment beneath central and east Asia, were likely responsible for the EECO.”

Lines 74-75: There is no widely accepted explanation for the PETM, but saying that there is no ‘viable’ explanation would be overly dismissive of the large body of work that has been done on the PETM, e.g., on large methane release, and magmatic intrusions into organic-rich sediments linked to the North Atlantic Large Igneous Provinces. Again, no references are provided to support this claim. Surprisingly, a discussion of potential triggers is included in the final section of the manuscript, which does not seem consistent with this strong statement.

We agree with the reviewer and have deleted this sentence, but also clarify in the text why the lack of a Hg signal in North Atlantic sediment cores place the NAIP as the trigger for the PETM in question.

Line 76: It reads now as if the question of what caused the PETM can be directly addressed by new age and geochemical data from the western Aleutian arc. I reiterate that it may be better to only mention in the Introduction that the climatic implications of the new results will be discussed later on in the article.

Have done.

Lines 78-83: Reads like a very short geological background. As mentioned above, this should be expanded to a summary that describes key geological and tectonic constraints on the evolution of the Aleutian subduction zone and arc.

Have done.

Lines 149-156: This is an important result, arguing against several previous models that linked the Komandorsky

Block to the Kronotsky arc. Are the geochemical signatures also different from the early Cenozoic Kronotsky arc rocks? I suggest showing in Fig. 3, or at least mentioning, the difference in geochemical signature.

Unfortunately, this is not as straight-forward as it may seem, since there are few samples for which age data and initial Sr-Nd-Pb isotope data sets are available. The ones we are aware of do not contain all parent-daughter elements necessary for making age corrections for radiogenic ingrowth. This is in particular true for the U-Th-Pb systems. Therefore, it is not possible to make such a comparison.

Line 152: the paleomagnetic data of Upper Cretaceous rocks attributed to the Kronotsky arc yield a paleolatitude of $\sim 30^\circ$ (Harbert et al., 2009 SMSPS), which I would not refer to as 'equatorial'.

Have changed.

Lines 161-163: Davis et al. (1989), cited here and in Fig. 1, obtained 54.4-50.2 Ma K-Ar ages from magmatic rocks of the Beringian margin, leading e.g. Scholl (2007) and Vaes et al. (2019) to assume a ~ 50 Ma jump of subduction to the incipient Aleutian subduction zone. Any ideas why magmatism would still be active there?

We agree with Vaes et al. (2019) that the proto-Kommandorsky Basin is likely to have opened early in the history of the Aleutian arc (between SZI and ~ 50 Ma). Therefore, as noted by these authors, continued subduction along the Beringian margin is likely to have compensated this opening. This is consistent with Aleutian initiation being primarily the result of cessation of subduction beneath the Olyutorsky Arc and subduction of the Izanagi-Pacific Ridge and not simply a jump from the Beringian margin to the Aleutians.

Lines 164-165: It is not clear why a very high convergence velocity of 10-20 cm/yr is assumed, based on the two old papers cited here. Why not derive this velocity from the GPlates plate model?

We did and it is consistent with a convergence velocity of ~ 20 cm/yr at 55 Ma between the Kula Plate and the overriding Aleutian/North American crust, now mentioned in the manuscript.

Lines 211-212: The spontaneous nature of the initiation of the IBM subduction zone is still debated (see e.g., Cramer et al. 2020 Nat. Comm., Liu et al. 2024 Comm. Earth & Env., van de Lagemaat et al. 2024 GR). Regardless of whether the authors support the spontaneous subduction initiation model, it would be fair to mention that it is still a topic of active research.

We now point out that Cramer et al. do not think that any subduction is purely spontaneous, even though many papers have referred to IBM subduction as being spontaneous. We also note a possible role of the Manus plume in triggering IBM SZI.

Line 217: The ages quoted here are not entirely correct if derived from Vaes et al. (2019). This paper, referenced here, estimated the initiation of the Olyutorsky arc at ~ 85 Ma and showed that the collision with Kamchatka, although indeed starting at 60-55 Ma, was diachronous, with obduction of its southern segment at ~ 55 -50 Ma and of its northern segment at ~ 50 -45 Ma. This is described further on in the manuscript (lines 261-264), so why not already mention that here?

Have done.

Lines 245-247 and Fig 6c: Why would the Pacific plate motion change to almost due north when the Izanagi-Pacific ridge just subducted and subduction of the actual Pacific plate below East Asia also initiated at around 56 Ma? It is not clear what is meant with the statement that the change (clockwise rotation?) in plate motion is also observed when keeping West Antarctica fixed. I assume the authors mean that the change is visible in the marine magnetic anomalies, but without showing that this remains difficult to follow.

Subduction beneath the Pacific Plate re-initiated at around 53 Ma in the revised GPlates model.

In the comments above, Reviewer 3 asks for separating relative plate motion changes from changes or uncertainties in the mantle reference frame. This is precisely why we refer to the clockwise change in Pacific Plate motion relative to a fixed West Antarctica. All relative plate motions are expressed as rotations between plate pairs, one fixed and one moving, so if one is interested in finding out where there has been a major change in Pacific relative plate motion based on the seafloor spreading record, the best way to do that is by using the Pacific-West Antarctic plate pair (West Antarctica being the only passive margin along the Pacific rim). The data that this is based on are all comprehensively documented in Wright et al. (2015) so there should be no reason to reproduce the data here. We now cite the Wright et al. (2015) paper in this sentence.

Lines 300-302 and 315-316: It is not straightforward to see how the slab pull of the Kronotsky subduction zone could have affected Pacific plate motion in a scenario where the Kula plate, and not the Pacific plate, subducts below the Kronotsky arc (Fig. 6). The youngest history of the Kronotsky arc discussed here is confusing: the authors mention the ~42 Ma estimate for the cessation of the arc, but at the same time refer to a 47 Ma slab break-off and cessation of northward-dipping subduction (Fig. 6f). I do not understand how there is a polarity reversal when subduction stops.

Please note that the slab breakoff and polarity reversal are a part of the Hu et al. (2022) model and do not play a big part in our interpretation in this version of the manuscript.

Figure 6 also does not show any southward-dipping subduction post-55 Ma anywhere in the northern Pacific.

The revised Figure 6 does, although as noted above, we do not think it significantly affects the plate motion change between 48-47 Ma.

Lines 320-321: Which hotspot reference frame?

Please see detailed summary of the plate motion reconstructions in the methods section.

Lines 327-330: It is clear that many major tectonic changes occurred in this time interval. Given that the authors focus a lot on the timing of these events, wouldn't it be helpful to place these sequential events in the framework of a tectonic chain reaction (Gürer et al., 2022 Nat. Geosc.)?

This goes beyond the scope of this manuscript, but would make a great follow-up study.

Data availability: The link to the Dryad repository did not work on my computer; it did not seem active. I advise to double-check this, as open access to the data is essential for future scrutiny of the data.

Thanks for pointing this out. We have now made sure that the link at Dryad is active. The model files and data are available via a Dropbox link for the review process, and will be uploaded to Zenodo with a DOI upon manuscript acceptance.

Response to REVIEWERS' COMMENTS

We are very grateful to all three reviewers for their very helpful, constructive and positive comments. We address their minor comments to their last review below.

Reviewer #1 (Remarks to the Author):

Review of Hoernle et al: Aleutian arc initiation at 56 Ma: Tectonic and climatic implications
The authors have sufficiently addressed my previous concerns. I provide a few additional comments/concerns that if addressed may strengthen the manuscript, wherever it is published. This is an important dataset which constrains the timing of initiation of the Western Aleutian Arc and informs regional Pacific plate tectonic models.

Lines 32-48 should be supported by references. Especially the discussion of previous geochronological constraints. Perhaps these are refs 1-8 as the next paragraph begins by citing reference 9?

As noted by the reviewer, references 1-8 were inadvertently left out and have been added to the text again.

The authors motivate the importance/impact of the new geochronology by emphasizing that previous ~55Ma ages are “unreliable” because they’ve not been reproduced since and because the uncertainties are large (lines 39-42). I do not think this necessarily makes them ‘unreliable’ does it? A date can be imprecise but still accurate. In the spirit of critical review, could one argue that these new results more precisely confirm, and are consistent with, previous constraints?

We have modified the text in line with the reviewer’s suggestion: “they are imprecise and provide limited temporal constraints on Aleutian initiation because of their large age uncertainties”

Line 74: HEB needs to be defined.

We now only use the HEB abbreviation in the caption of figure 5 and define it there.

Line 78: Why the TF and FZ acronyms?

We have replaced the abbreviations (acronyms) with transform fault and fracture zone, because we do not use these terms enough to justify abbreviating them.

Lines 552-554 need citation(s)

This sentence was based on our calculation. Since the amount of new subduction zone cannot be constrained well, we have deleted this sentence.

I like the addition of Figure 4. Expanding panels e-h to show Beringian Margin subduction until ~50 Ma (Davis et al., 1989) would be more comprehensive and better support (somewhat) the authors’ description/interpretation of their preferred model for western Aleutian initiation (as well as text in lines 327-329).

Have expanded panels e-h to show Beringian Margin subduction until ~50 Ma.

The manuscript needs to be checked throughout for use, placement, and defining of acronyms as there are instances where they are not defined, redefined, and/or perhaps not necessary.

We checked the manuscript. As indicated above, we have removed unnecessary abbreviations (acronyms).

Reviewer #2 (Remarks to the Author):

Thank you for the thorough and thoughtful revisions made to your manuscript in response to the previous round of review. I have carefully examined the revised version and your detailed point-by-point responses. I am pleased to note that you have addressed all of my major concerns comprehensively and effectively.

The modifications to the text, the additional data included (particularly regarding comparison of Izu-Bonin-Mariana (IBM) with Aleutian and Kamchatka subduction/arc initiation), and the refined interpretations have significantly strengthened the manuscript. The arguments are now more robust, the conclusions are well-supported by the evidence presented, and the overall clarity has been improved.

In its current form, the work presents a noteworthy contribution to the field through rare samples from lavas and granodiorites from four submarine basement sequences spanning the western Aleutian Arc. The methodology is sound, the results are compelling, and the discussion appropriately contextualizes the findings within the established literature.

I believe the manuscript now meets the high standards expected for publication in Nature Communications. I have no further substantive criticisms and recommend its acceptance without further revision.

I congratulate the authors on a fine piece of work.

Best regards,
Zhao Lei

Reviewer #3 (Remarks to the Author):

First, I would like to compliment the authors for their thorough revisions and detailed responses to all points raised by the reviewers. In my opinion, the decision to focus the manuscript on the tectonic implications of their new data, rather than speculative climatic impacts, has significantly strengthened its scientific value. The revised manuscript represents a clear and important advance in our understanding of the complex tectonic history of the northern Pacific region. By integrating recent tectonic models with their new geochemical and age constraints, the authors provide a robust and comprehensive model for the initiation and early evolution of the Aleutian subduction zone and arc. Moreover, the addition of Figures 4 and 5 provide strong visual support for the tectonic model and interpretations of the data. The authors now also discuss enigmatic features of this region, like the Shirshov and Bowers Ridges, in more detail and highlight some outstanding questions and problems for future research. I consider this new tectonic model for the Aleutian arc and broader Bering Sea region as the state-of-the-art and am confident it will serve as a platform for further research. Finally, the authors provide a more nuanced discussion of the potential links between the Aleutian arc initiation and climatic events like the PETM, carefully outlining possible mechanisms. I therefore recommend this manuscript for publication in Nature Communications, after addressing the (very) minor revisions suggested below.

Title: It may be good to indicate the uncertainty in the age of arc initiation, e.g., by writing '~56 Ma'.

Agree. Have done so.

Introduction: the tectonic models discussed here built on important work of Russian geologists. I

realize that the number of references is limited but citing some key works on arc-continent collision in Kamchatka like Konstantinovskaya (2001, Tectonophys.) and Shapiro & Soloviev (2009, Russ. Geol. Geophys.) would help acknowledging this.

We have cited the works on which we base our discussion and are already over the limit of 70 references in the main body of the manuscript, therefore, we have not added these and other possible citations.

Lines 32-48: The references to published ages (1-8) seem to be missing in the revised manuscript.

Have added these.

Line 175: 'was not accreted until the mid-Miocene'

We have deleted "presumably" as recommended by the reviewer.

Caption Figure 4: Add that Manus plume 'may have caused uplift'?

Have done so.

Line 476: replace 'no doubt' by 'undoubtedly'

Have done.

Lines 548-551: This is a very long and complicated sentence, I would suggest splitting it up and rewriting it.

Agreed, have broken into two sentences.

Bram Vaes